# Chemical design of electronic and magnetic energy scales of tetravalent praseodymium materials

Arun Ramanathan[1], Jensen Kaplan[2], Dumitru-Claudiu Sergentu [3,4,5], Jacob A. Branson [6,7], Mykhaylo Ozerov [8], Alexander I. Kolesnikov [9], Stefan G. Minasian [7], Jochen Autschbach [5], John W. Freeland [10], Zhigang Jiang [2], Martin Mourigal [2] & Henry S. La Pierre [1,11]✉

Lanthanides in the trivalent oxidation state are typically described using an ionic picture that leads to localized magnetic moments. The hierarchical energy scales associated with trivalent lanthanides produce desirable properties for e.g., molecular magnetism, quantum materials, and quantum transduction. Here, we show that this traditional ionic paradigm breaks down for praseodymium in the tetravalent oxidation state. Synthetic, spectroscopic, and theoretical tools deployed on several solid-state $Pr^{4+}$-oxides uncover the unusual participation of $4f$ orbitals in bonding and the anomalous hybridization of the $4f^1$ configuration with ligand valence electrons, analogous to transition metals. The competition between crystal-field and spin-orbit-coupling interactions fundamentally transforms the spin-orbital magnetism of $Pr^{4+}$, which departs from the $J_{eff} = 1/2$ limit and resembles that of high-valent actinides. Our results show that $Pr^{4+}$ ions are in a class on their own, where the hierarchy of single-ion energy scales can be tailored to explore new correlated phenomena in quantum materials.

The electronic structure of lanthanide and actinide materials inherits on-site correlations and unquenched orbital degrees of freedom from atomic $f$-electron states. In the most stable trivalent oxidation state ($Ln^{3+}$), the core-like $4f$ orbitals are usually only weakly perturbed by the crystalline environment[1]. Yet, the energetic splitting of the otherwise $2J + 1$ fold-degenerate (free-ion) ground-state yields rich physics and applications. For example, $Ln^{3+}$ insulators can host anisotropic magnetic moments with effective spin-1/2 character ($J_{eff} = 1/2$) that are promising to stabilize entangled states in quantum magnets[2,3]. Metallic $4f$ and $5f$ systems also display a wealth of quantum phenomena rooted

in the hybridization between localized $f$-electrons and conduction $d$-electrons such as the Kondo effect, valence fluctuations, correlated insulators, and unconventional superconductivity[4,5].

In rare instances, Ce, Pr, Tb, (and less definitively Nd, and Dy) ions exist in a high, formally tetravalent, oxidation state, i.e., $Ce^{4+}$ ($4f^0$), $Pr^{4+}$ ($4f^1$), and $Tb^{4+}$ ($4f^7$)[6,7]. Although covalent metal-ligand interactions involving the $4f$ shell are generally weak in $Ln^{3+}$ systems[8,9], this paradigm breaks down for $Ln^{4+}$ as $4f$ orbitals directly participate in bonding and hybridize with the valence orbitals of the ligands (e.g., the $2p$ states for oxygen) analogous to transition metals. A high oxidation state and

[1]School of Chemistry and Biochemistry, Georgia Institute of Technology, Atlanta, GA 30332, USA. [2]School of Physics, Georgia Institute of Technology, Atlanta, GA 30332, USA. [3]University of Rennes, CNRS ISCR (Institut des Sciences Chimiques de Rennes) - UMR 6226, F-35000 Rennes, France. [4]A. I. Cuza University of Iași, RA-03 Laboratory (RECENT AIR), Iași 700506, Romania. [5]Department of Chemistry, University at Buffalo, State University of New York, Buffalo, NY 14260-3000, USA. [6]Department of Chemistry, University of California, Berkeley, CA 94720, USA. [7]Lawrence Berkeley National Laboratory, Berkeley, CA 94720, USA. [8]National High Magnetic Field Laboratory, Florida State University, Tallahassee, FL 32310, USA. [9]Neutron Scattering Division, Oak Ridge National Laboratory, Oak Ridge, TN 37831, USA. [10]Advanced Photon Source, Argonne National Laboratory, Lemont, IL 60439, USA. [11]Nuclear and Radiological Engineering and Medical Physics Program, School of Mechanical Engineering, Georgia Institute of Technology, Atlanta, GA 30332, USA. ✉e-mail: hsl@gatech.edu

strong $4f$ covalency are expected to significantly impact the redox, electronic, and magnetic properties of these systems, but, surprisingly, only a few $Ln^{4+}$ insulators have been studied in detail to date[10,11]. As $Ce^{4+}$ is nominally non-magnetic and $Tb^{4+}$ has a half-filled $4f$ shell, the one-electron $4f^1$ configuration of $Pr^{4+}$ makes it unique to search for new quantum phenomena at the nexus of strong electronic correlations, quantum magnetism, and spin-orbital entanglement.

The emergence of an insulating state in $PrBa_2Cu_3O_{6+\delta}$ (PBCO)—a compound obtained by substituting Y by Pr in the high-$T_c$ superconductor $YBa_2Cu_3O_{6+\delta}$ (YBCO), and valence fluctuations driven metal-insulator transitions in Pr-containing complex oxides—epitomizes such anomalous behavior. In PBCO, the significant Pr-$4f$/O-$2p$ covalency (Fehrenbacher-Rice hybridization) drives a mixed-valent state for Pr ions that competes with Cu-$3d$/O-$2p$ hybridization (Zhang-Rice) and dramatically suppresses superconductivity in favor of local magnetism[12]. In Pr containing complex oxides like $(Pr_{1-y}Y_y)_{1-x}Ca_xCoO_{3-\delta}$ and $Pr_{1-x}Sr_xCoO_3$, valence transition from $Pr^{4+}$ to $Pr^{3+}$ drives a spin state/metal-insulator transition, making them attractive for oxide-based electronics[13-15]. This observation stimulated early experimental and theoretical work to understand the magnetism of cubic $Pr^{4+}$ oxides such as $PrO_2$ and $BaPrO_3$[10,11]. More recently, the edge-sharing $PrO_6$ octahedral in $Na_2PrO_3$ have attracted attention to stabilize antiferromagnetic Kitaev interactions between $J_{eff}=1/2$ moments[16]. But much like in $PrO_2$[11], the hallmark of $Na_2PrO_3$ magnetism is the unusually large crystal field (CF) splitting $\Delta_{CF} \geq 230$ meV that competes with spin-orbit coupling (SOC) $\zeta_{SOC} \approx 100$ meV[17]. The competition between CF and SOC yields drastically different single-ion and exchange properties than expected in the $\Delta_{CF} \ll \zeta_{SOC}$ limit where $J_{eff}=1/2$ moments usually form, as illustrated in Fig. 1a. The most noticeable consequences for $Na_2PrO_3$ are the low effective magnetic moment with $g \approx 1$ and the surprisingly large $J_{ex} \approx 1$ meV exchange interactions[17]. The precise mechanisms that endow $Pr^{4+}$ ions with these unusual properties are poorly understood.

In this work, we focus on the microscopic mechanisms that underpin the electronic and magnetic behavior of $Pr^{4+}$ materials comprising octahedral $[PrO_6]^{8-}$ units. We examine a series of insulating oxides with decreasing order of lattice dimensionality: quasi-2D layers in $Na_2PrO_3$ (**2-Pr**, Fig. 1b, ref. 18) and quasi-1D chains in $Sr_2PrO_4$ (**1-Pr**, Fig. 1c, ref. 19) to understand magnetic exchange and the role of Pr-$4f$/O-$2p$ hybridization, and quasi-isolated "0D" $PrO_6$ octahedra in $Li_8PrO_6$ (**0-Pr**, Fig. 1d, ref. 20) to understand the intrinsic behavior of the $[PrO_6]^{8-}$ moiety without the complication of magnetic exchange interactions. Using inelastic neutron scattering (INS) and infrared magnetospectroscopy (IRMS), we demonstrate that the magnetic ground-state of $Pr^{4+}$ ions systematically deviates from the $J_{eff}=1/2$ limit and can be understood from an intermediate coupling regime where significant admixture of nominally excited $J$-states enter the ground-state wavefunction. X-ray magnetic circular dichroism (XMCD) at the Pr-$M_{4,5}$ edge strengthens this picture and elucidates the mechanism behind the low effective magnetic moments of $Pr^{4+}$ ions. Oxygen $K$-edge and Pr $M_{4,5}$-edge x-ray absorption spectroscopy (XAS) evidences Pr-$4f$/O-$2p$ hybridization with a degree of Pr-O covalency influenced by the symmetry of the $[PrO_6]^{8-}$ moiety. These results are supported and explained by first-principles calculations. Taken together, this study establishes $Pr^{4+}$ ions as an important building block to design quantum magnetic behavior distinct from trivalent lanthanides. Furthermore, the study demonstrates that the competition between energy scales in the $[PrO_6]^{8-}$ unit is reminiscent of $4d$ and $5d$ transition metals[21], and can serve as a simplification of $5f^1$ actinide systems for which $\Delta_{CF}$, $\zeta_{SOC}$ and on-site Hubbard interaction $U$ strongly compete[22,23].

## Results

Crystalline powder samples of **0-Pr, 1-Pr**, and **2-Pr** were synthesized using solid-state reactions and phase purity was confirmed using powder X-ray diffraction (See Methods and Supplementary Methods 1, 2). These materials incorporate low symmetry, but close-to-octahedral $[PrO_6]^{8-}$ units with $D_{2d}$ symmetry in **2-Pr** ($C_{2/c}$ space group, Fig. 1b), $C_{2h}$ in **1-Pr** ($Pbam$ space group, Fig. 1c), and $S_6$ in **0-Pr** ($R\bar{3}$ space group, Fig. 1d). **0-Pr** contains spatially isolated $PrO_6$ octahedral with the nearest-neighbor Pr-Pr distance of $d \approx 5.6$ Å, which is significantly longer than $d \approx 3.5$ Å in **1-Pr** and **2-Pr**, and sufficient to effectively magnetically isolate the $[PrO_6]^{8-}$ units.

Broadband INS measurements were used to probe the CF states accessible to the dipole selection rule. Given that $Pr^{4+}$ is a $4f^1$ Kramers ion (isoelectronic to $Ce^{3+}$), the standard approach starts from a $^2F$ free-ion manifold split by SOC into $J=5/2$ and $J=7/2$ multiplets. For a six-oxygen environment with $O_h$ symmetry, the CF further splits the $^2F_{5/2}$ multiplet into a doublet ground-state ($\Gamma_7$) and an excited quartet ($\Gamma_8$), and the $^2F_{7/2}$ multiplet into two doublets ($\Gamma_7'$ and $\Gamma_6$) and a quartet ($\Gamma_8'$). Any deviation from $O_h$ symmetry, as is the case for our materials (see Fig. 1), splits the quartets and yields seven Kramers doublets (KDs). Thus, the magnetic properties of $Pr^{4+}$ ions in the hypothetical $\Delta_{CF} \ll \zeta_{SOC}$ limit are dominated by the $\Gamma_7$ doublet ground-state, which is spanned by pseudospin variable $|\pm\rangle$ associated with an effective angular momentum $J_{eff}=1/2$. The wave function of the $\Gamma_7$ doublet is well-known[16] and can be written in either $|J,m_J\rangle$ or $|m_l,m_s\rangle$ basis (see Fig. 1a and SI).

However, as $\zeta_{SOC} \approx \Delta_{CF}$ for $Pr^{4+}$, the simple $J_{eff}=1/2$ picture breaks down. Indeed, INS on **0-Pr** readily reveals an intense magnetic signal at $E_1^{0-Pr}=274(1)$ meV which we assign to the *first* CF excitation, see Fig. 2a. This energy is 2.5 times larger than reported for isoelectronic $Ce^{3+}$ in $KCeO_2$[24,25], and to the best of our knowledge, this is the largest first CF excited state observed for a lanthanide ion. Given the uncommonly large $\Delta_{CF}$, modeling the single-ion properties of **0-Pr** requires an intermediate coupling approach[11] that uses the complete set of 14 $|m_l,m_s\rangle$ basis states and diagonalizes the single-ion CF Hamiltonian $\hat{\mathcal{H}}_{CF}^{0-Pr} = B_2^0\hat{O}_2^0 + B_4^0\hat{O}_4^0 + B_4^4\hat{O}_4^4 + B_6^0\hat{O}_6^0 + B_6^4\hat{O}_6^4$ for a fixed value of the spin-orbit interaction $\zeta_{SOC}$ (see Methods and Supplementary Note 1 for definitions). The above single-ion CF Hamiltonian is written in a *truncated* symmetry basis to avoid over-parametrization and treat all materials on equal footing (see Methods). Irrespective, it is impossible to constrain the parameters of $\hat{\mathcal{H}}_{CF}^{0-Pr}$ solely using the one observed excitation. Thus **1-Pr** is examined first because the $PrO_6$ octahedra further depart from ideal symmetry and is likely to present a richer spectrum in INS.

Unlike **0-Pr**, INS results on **1-Pr** reveal three magnetic excitations at $E_1^{1-Pr}=168(1)$ meV, $E_2^{1-Pr}=335(1)$ meV, and $E_3^{1-Pr}=387(1)$ meV (Fig. 2d); more states than available in the sole $J=5/2$ manifold. Although **1-Pr** exhibits an antiferromagnetic transition at $T_N=3.0$ K with a pronounced peak in $\chi(T)$ (Fig. 2e), the magnetic susceptibility at $\mu_0H=3$ T above $T>40$ K can be used to further constrain the parameters of the single-ion CF Hamiltonian. To proceed, $\hat{\mathcal{H}}_{CF}^{Pr}$ is diagonalized with fixed $\zeta_{SOC}=112$ meV (free-ion value) and the CF parameters fit to the observed INS energies and magnetic susceptibility data (see Methods and Supplementary Note 1). This yields a model Hamiltonian that reproduces the isothermal magnetization at $T=50$ K (Fig. 2f) and predicts an unusually small powder-averaged $g$ factor $g_{CF}^{avg,1-Pr}=1.13$ and an effective moment $\mu_{CF}^{eff,1-Pr} \approx 1\mu_B$/Pr comparable to the value extracted from a Curie–Weiss fit $\mu_{CW}^{eff,1-Pr}=1.13(1)\mu_B$/Pr (Fig. 2e).

Having established an approach to model the single-ion Hamiltonian for $Pr^{4+}$, **0-Pr** is examined and employ IRMS measurements conducted up to 17.5 T. The normalized IR spectra reveal three field-dependent features around $E_1^{0-Pr}=267$ meV, $E_2^{0-Pr}=270$ meV, and $E_4^{0-Pr}=670$ meV (Fig. 2c) that can be associated with magnetic dipole allowed CF transitions from the ground-state doublet. The distinct features at $E_1^{0-Pr}$ and $E_2^{0-Pr}$—resolved due to the excellent energy resolution of IRMS—correspond to the sole transition observed in INS. The first excited level in **0-Pr** is thus a quasi-degenerate quartet

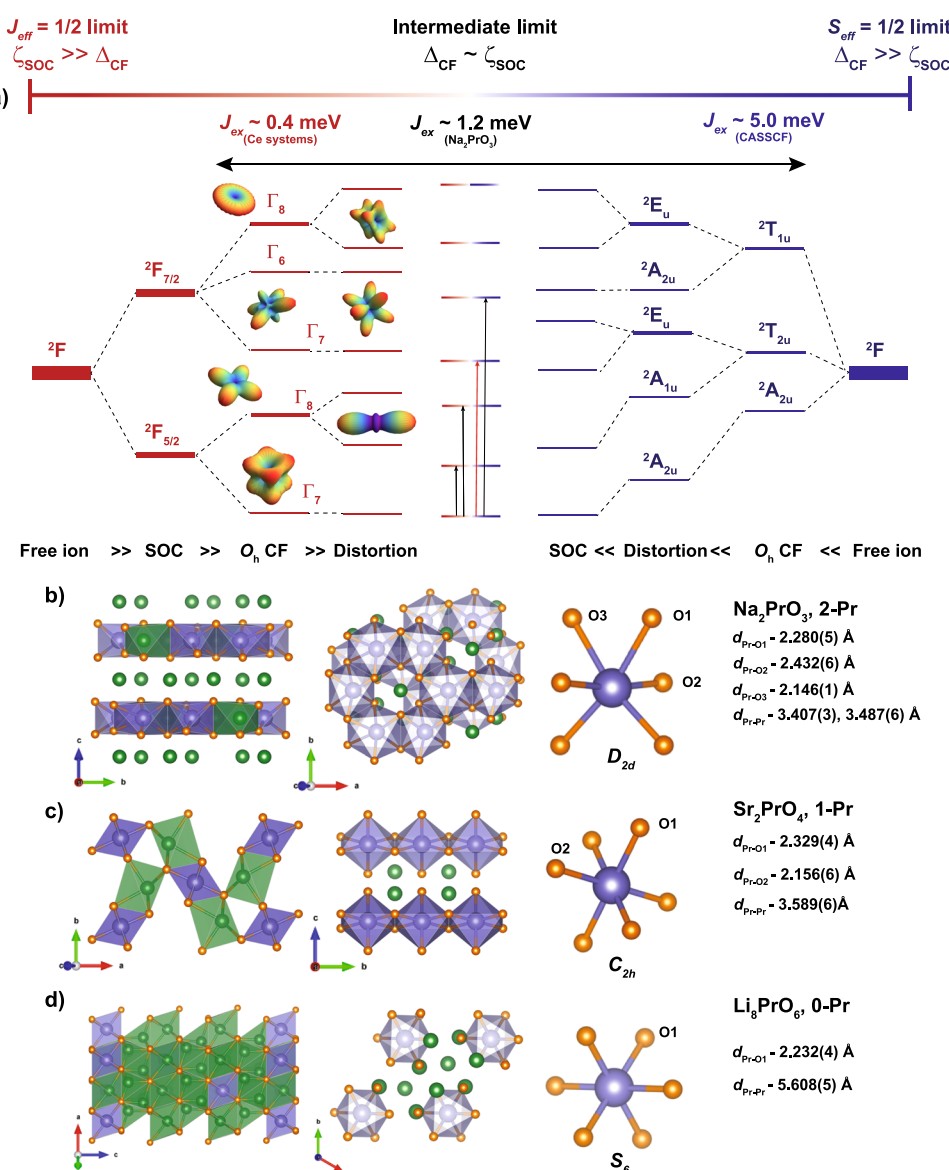

**Fig. 1 | Competing interactions in Pr$^{4+}$ oxides. a** Hierarchy of single-ion energy scales for Pr$^{4+}$ ions in octahedral oxygen environments starting from a $^2F$, ($S = 1/2$, $L = 3$) free-ion state. For spin-orbit coupling (SOC) as the dominant energy scale ($J_{eff} = 1/2$ limit, left, brown), the low-symmetry crystal field (CF) lifts the degeneracy of the ground-state $^2F_{5/2}$ and excited $^2F_{7/2}$ multiplets into seven Kramers doublets (KDs, with selected radial squared wave-functions represented). For approximate $O_h$ symmetry, the $\Gamma_7$ doublet, in the $|J,m_J\rangle$ basis, is given by $\Gamma_7^\pm = \sin\theta|\tfrac{5}{2}, \pm\tfrac{5}{2}\rangle + \cos\theta|\tfrac{5}{2}, \mp\tfrac{3}{2}\rangle$, where $\sin^2\theta \approx 1/6$ (See SI). For CF as the dominant energy scale ($S_{eff} = 1/2$ limit, right, blue), SOC and distortion from $O_h$ symmetry lift the $^2A_{2u}$ ground state and the triply degenerate $^2T_{2u}$ and $^2T_{1u}$ excited

states into seven KDs. The ground-state doublet is given in the $|m_l, m_s\rangle$ basis by $|\pm\rangle = A|\mp3, \pm\tfrac{1}{2}\rangle - B|\mp2, \mp\tfrac{1}{2}\rangle + C|\pm1, \pm\tfrac{1}{2}\rangle - D|\pm2, \mp\tfrac{1}{2}\rangle$, where $(A^2/B^2)^{\Gamma_7} \approx 2.6$ and $(C^2/D^2)^{\Gamma_7} \approx 0.33$ for the $|\Gamma_7\rangle$ doublet[62] (See SI). In the intermediate limit, the competition between $\Delta_{CF}$ and $\zeta_{SOC}$ scales in Pr$^{4+}$ yields seven KD (indicated by the mix of brown/blue lines) with magnetic properties that are distinct from the $J_{eff} = 1/2$ and $S_{eff} = 1/2$ limits, such as large magnetic super-exchange achievable by tuning the ligand field. **b−d** Crystal structure, magnetic lattice dimensionality, and Pr$^{4+}$ coordination environment for the oxides studied in this work: Na$_2$PrO$_3$ (**2-Pr**), Sr$_2$PrO$_4$ (**1-Pr**), and Li$_8$PrO$_6$ (**0-Pr**), respectively.

($\Gamma_8$-like) split by the weak distortion of the PrO$_6$ octahedra from an ideal $O_h$ symmetry. This model is fully supported by ab-initio calculations (multireference CASPT2 + SOC, Methods and Supplementary Note 2), which predicts the quasi-degeneracy of $E_2^{0-Pr}$ and $E_2^{0-Pr}$ at 241 and 246 meV, respectively. The 670 meV transition is likely weak in INS and masked by the strong background (recoil intensity observed from hydroxide impurities, <5% in starting materials, see Supplementary Methods 1). The ab-initio calculations assign it as the fourth ground to excited state transition, and further predict a third (IR inactive)[26] transition at 396 meV with $^2T_{2u} + ^2A_{2u}$ origin. The parameters of $\hat{\mathcal{H}}_{CF}^{1-Pr}$ are fitted using the same procedure as for **0-Pr** (Fig. 2b). The resulting model yields $g_{CF}^{av,0-Pr} = 0.94$, in good agreement with the isothermal

magnetization (Fig. 2f), and $\mu_{CF}^{eff,0-Pr} = 0.81\mu_B$/Pr consistent with $\mu_{CW}^{eff,0-Pr} = 0.86(1)\mu_B$/Pr (Fig. 2b) and first-principles calculations (See Discussion). Analysis of the INS spectrum of **2-Pr** leads to similar conclusions[17].

Analysis of the single-ion physics of these three materials therefore suggests that the ground-state wavefunction of Pr$^{4+}$ systematically deviates from the $\Gamma_7$ doublet expected in the $J_{eff} = 1/2$ limit. For example, the ratio of $|m_l = \mp3, m_s = \pm1/2\rangle$ to $|m_l = \pm2, m_s = \pm1/2\rangle$ basis states (parametrized by $(A^2/B^2)$, see Fig. 1 caption and Methods for Definition and Supplementary Tables 5−7 for full wavefunction) changes from 2.6 for the $|\Gamma_7\rangle$ doublet to 0.53, 2.13, and 1.51 for the ground-state doublet of **0-Pr**, **1-Pr**, and **2-Pr**, respectively. When recast

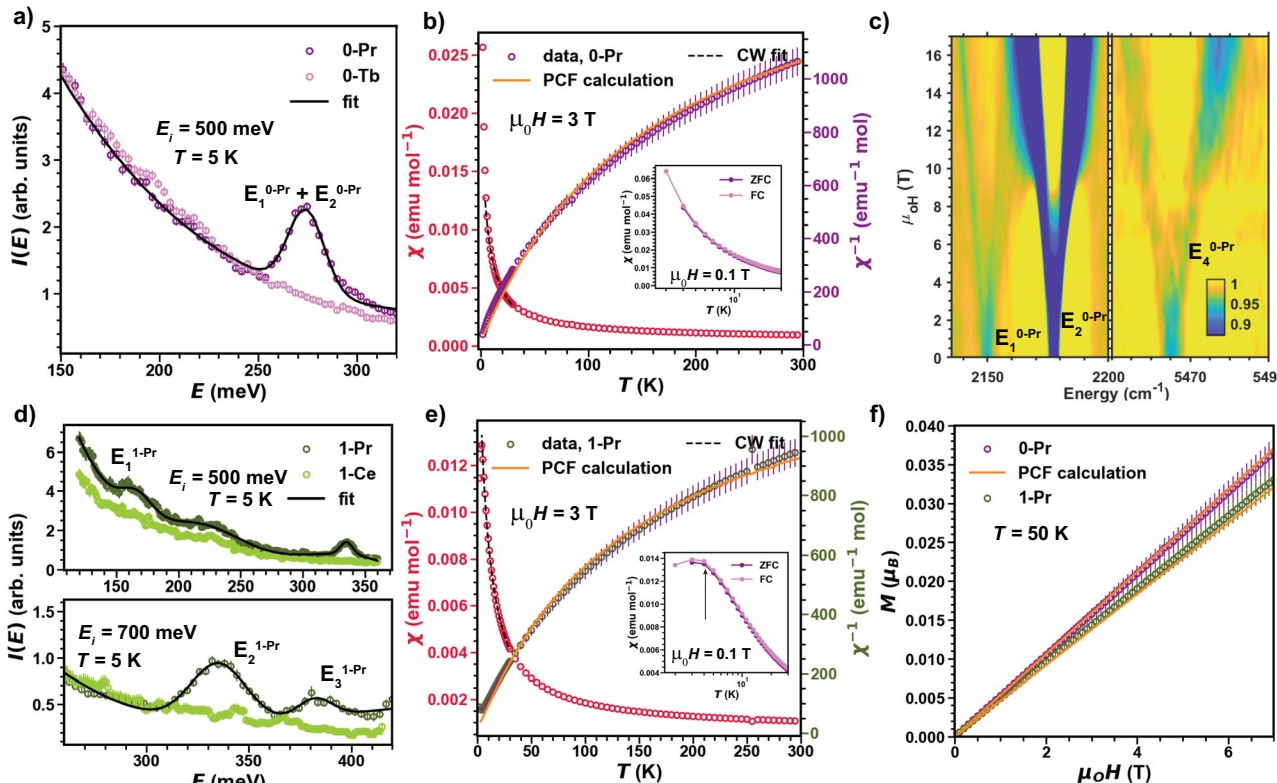

**Fig. 2 | Crystal field excitations and magnetic properties in Pr⁴⁺ oxides.**
**a, d** Energy-dependent neutron-scattering intensity at low temperature integrated into the range $6 < Q < 7$ Å$^{-1}$ for several neutron energies and for **1-Pr** (Sr$_2$PrO$_4$) and **0-Pr** (Li$_8$PrO$_6$), respectively. **b, e** Magnetic susceptibility ($\chi(T)$) and inverse susceptibility ($\chi(T)^{-1}$) data measured under $\mu_0 H = 3$ T. Also plotted is a CF model and a Curie−Weiss analysis in the temperature range $4 < T < 40$ K that yields $\Theta_{CW}^{1\text{-}Pr} = -7.52(2)$ K. The CF model calculations were carried out in Stevens operator formalism using the PYCRYSTALFIELD package[58] with 14 $|m_l, m_s\rangle$ basis states to account for intermediate coupling. The inset shows macroscopic magnetic

behavior under an applied field of $\mu_0 H = 0.1$ T. **1-Pr** exhibits an AFM order with a pronounced peak in $\chi(T)$ with no splitting between ZFC and FC. **c** Normalized IR transmission spectra as a function of applied magnetic field for **0-Pr**. The blue color indicates the area with intense CF transitions, while yellow corresponds to the flat line. The experimental data were taken at 5 K and normalized to the average spectra as described in Methods. **f** Isothermal magnetization $M(H)$ at $T = 50$ K for **0-Pr** and **1-Pr** plotted together with the CF model. $T = 50$ K was chosen so that **1-Pr** is well above the ordering temperature and free from short-range correlations.

in the $|J, m_J\rangle$ basis, our analysis suggests that intermediate coupling mixes $|J = 7/2, m_J = \pm 3/2\rangle$ and $|J = 7/2, m_J = \pm 5/2\rangle$ states into the ground doublet, leading to an increased $|J, m_J = \pm 5/2\rangle$ character for the ground-state wavefunction (see Supplementary Note 1).

Probing the density of electronic states around the $4f$ level, XAS (see Methods) provides definitive spectroscopic evidence of this hypothesis and further elucidates the origin of the large $\Delta_{CF}$. The $M_5$ ($3d_{5/2} \rightarrow 4f_{7/2}$ and $4f_{5/2}$) and $M_4$ ($3d_{3/2} \rightarrow 4f_{5/2}$) edges for **2-Pr** and **0-Pr** are shown in Fig. 3a, b. The splitting between the Pr $M_5$ edge at 931 eV and $M_4$ edge at 951 eV originates from SOC within the $3d$ core-holes. The $M_{5,4}$ edges for both **0-Pr** and **2-Pr** show intense main peaks (labeled $A$ and $B$) followed by higher-energy satellites ($A'$ and $B'$) raised by ≈5 eV, and a smeared shoulder (labeled $B^S$) starting ≈3 eV below the main peaks. For a $4f^1$ ion with a ground-state wavefunction described by a pure $J$-multiplet, isotropic $M_{5,4}$ edges are expected[27] as predicted by the Wigner−Eckart theorem (Fig. S4). Thus, the complex spectral lineshapes observed in **0-Pr** and **2-Pr**, which resemble previous observations for PrO$_2$[28] (see Supplementary Figs. 4, 5), are direct evidence for the mixing of $^2F_{7/2}$ and $^2F_{5/2}$ multiplets in the ground and excited states of our compounds. This analysis is corroborated by the branching ratio (BR) $I_{M_5}/(I_{M_5} + I_{M_4})$, evaluated from the total spectral weight under all $M_5$ or $M_4$ peaks, with values of 0.445(17) (**2-Pr**), 0.443(10) (**0-Pr**), and 0.453(6) (PrO$_2$). These values are less than ≈0.50 reported for ionic Ce$^{3+}$ systems[27,28] (a BR of ≈0.5 also applies for an ionic Pr$^{4+}$ system with no hybridization, as shown in Fig. S4).

The complex $M_{5,4}$ lineshapes and multiplet mixing in $4f^1$ systems have previously been ascribed to electronic hybridization, i.e., covalent bonding, between $4f$ and O-$2p$ states[27,29]. In the charge-transfer limit (Zaanen−Sawatzky−Allen (ZSA) scheme[30]), the electronic ground-state of Pr$^{4+}$ is a superposition $|\psi_g\rangle = \sin\theta |4f^1\rangle + \cos\theta |4f^2\underline{\nu}\rangle$ where $\underline{\nu}$ is a hole in the O $2p$ band. Configuration-interaction (CI) calculations using the Anderson-impurity model (AIM)[31,32] were carried out to understand the impact of $4f$ covalency on the $M_{5,4}$ XAS spectra of the subject compounds. Within this framework, the initial and final states include the combination of $|3d^{10}4f^1\rangle$ & $|3d^{10}4f^2\underline{\nu}\rangle$, and $|3d^94f^2\rangle$ & $|3d^94f^3\underline{\nu}\rangle$, respectively (see Methods). In the limit of vanishing hybridization ($V \rightarrow 0$), the energy difference between initial configurations is $\Delta E_g = 2.0$ eV (**2-Pr**) and 3.0 eV (**0-Pr**), and between final states is $\Delta E_f = 0.5$ eV (**2-Pr**) and 1.5 eV (**0-Pr**). Calculations for a realistic hybridization agree well with the experimental data (Fig. 3a, b) and allows to estimate the fraction of $4f^1$ and $4f^2$ configurations in $|\psi_g\rangle$ to be 69−31% for **2-Pr**, and 75−25% in **0-Pr**; i.e., **0-Pr** is the least hybridized system. The estimated contributions to $|\psi_g\rangle$ in **2-Pr** are similar to the values extracted for PrO$_2$ (see Fig. S5)[28]. The smaller hybridization in **0-Pr** relative to both **2-Pr** and PrO$_2$ is evident theoretically from the increased $\Delta E_g$ and experimentally from the more dominant main peak at both edges. Due to the comparable energy scales of $\Delta E_g \approx V_g$, Pr$^{4+}$ oxide systems are thus strongly correlated insulators in the charge-transfer limit ($U_{ff} \gg \Delta E_g$) and require a quantum many-body description for the ground state. Indeed, these systems behave similar to CeO$_2$

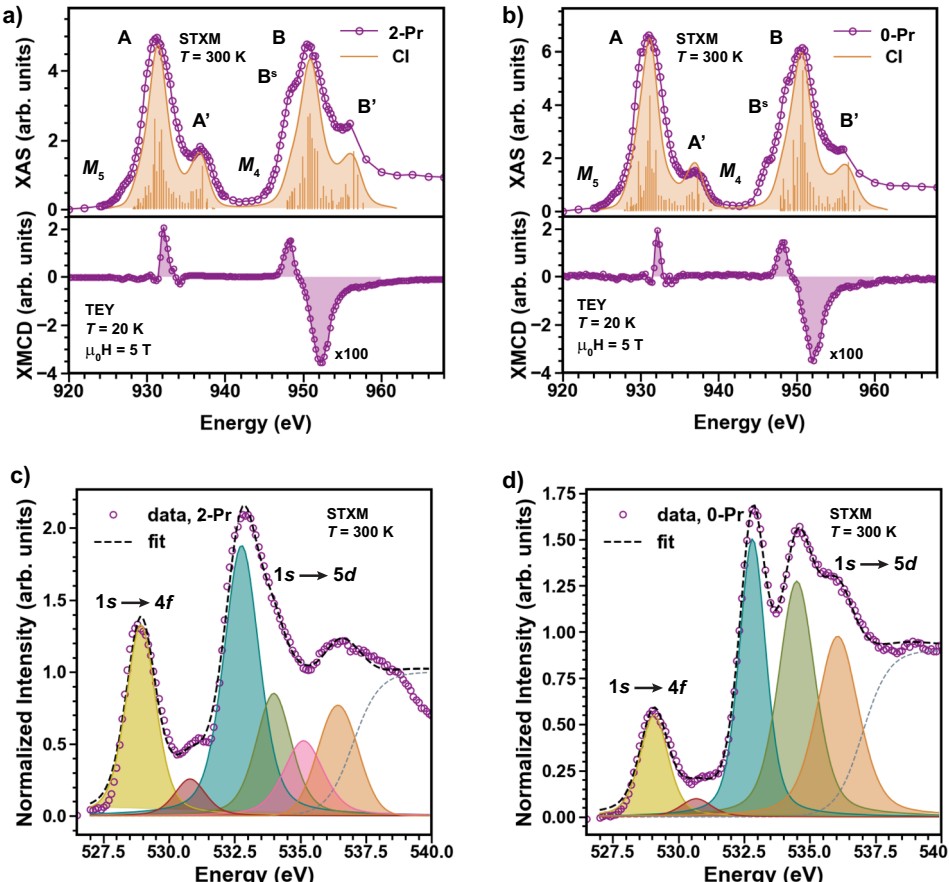

**Fig. 3 | Fingerprints of Pr-4f/O-2p hybridization from X-ray scattering spectra.**
**a, b** Isotropic XAS (top) and XMCD (bottom) spectra at the Pr $M_5$ and $M_4$ edges for
**2-Pr** (left) and **0-Pr** (right), respectively measured using the Scanning Transmission
X-Ray Microscope (STXM) mode (XAS, $\mu_0 H = 0$ T and $T = 300$ K) and the total
electron tield (TEY) mode (XMCD, $\mu_0 H = 5$ T and $T = 20$ K). For the XAS spectra, first-
principle calculations (CI under AIM framework) are shown as orange sticks with

Gaussian and Lorentzian broadening. For the XMCD spectra, the integration range
for the sum-rule analysis is shown as a purple shaded region. **c, d** Isotropic XAS
spectra at the oxygen $K$ edge for **2-Pr** (left) and **0-Pr** (right), both measured in STXM
mode ($T = 300$ K). The peak corresponding to Pr-4f/O-2p hybridization is shown in
yellow. For comparison, reference data on PrO$_2$[28] is shown in Table. S2 and Fig. S4
and S8.

and the spectral features are describing ground and excited state
charge transfer[33].

The weak magnetic moment observed for the Pr$^{4+}$ ion can also be
understood directly from XAS. The XMCD spectra at the Pr $M_{5,4}$ edge
(See Methods) reveal sizeable dichroism (Fig. 3a (**2-Pr**) and 3b (**0-Pr**)).
Quantitative analysis using sum rules (see Supplementary Note 1)
allows for the extraction of the orbital $\mu_o = -\langle L_z \rangle \mu_B$, spin ($\mu_s = -2\langle S_z \rangle \mu_B$,
and magnetic dipole $\langle T_z \rangle$ contributions to the total moment $\mu_t$[34] (see
Methods, note that $\langle T_z \rangle \neq 0$ reflects the departure from spherical
symmetry for $\mu_s$)). Applying the orbital sum-rule yields $\mu_o = 0.34(5)\mu_B$
(**2-Pr**) and $0.33(7)\mu_B$ (**0-Pr**). The measured absolute macroscopic
magnetization yields $\mu_t = 0.028(4)\mu_B$ (**2-Pr**) and $0.044(3)\mu_B$ (**0-Pr**) at
$\mu_0 H = 5$ T and $T = 20$ K, and in turn yields $\mu_s = -0.31(9)\mu_B$ (**2-Pr**) and $-$
$0.29(10)\mu_B$ (**0-Pr**) based on $\mu_{total} = \mu_{spin} + \mu_{orbital}$. These values corre-
spond to $|\langle T_z \rangle / \langle S_z \rangle| = 0.40(8)$ (**2-Pr**) and $0.38(1)$ (**0-Pr**) and $|\langle L_z \rangle /$
$\langle S_z \rangle| = 2.17(6)$ (**2-Pr**) and $2.29(9)$ (**0-Pr**); the latter is significantly lower
than $|\langle L_z^{4f^1} \rangle / \langle S_z^{4f^1} \rangle| = 8$ expected for a free $4f^1$ ion[35] but resembles $5f-c$
hybridized uranium systems[36,37] and is usually attributed to $4f$ electron
delocalization in lanthanides. Despite a low bulk magnetic moment,
XMCD data reveals the existence of sizable spin and orbital moments
with a reduced $|\langle L_z \rangle / \langle S_z \rangle|$ ratio that provides a fingerprint for Pr-4f/O-2p
hybridization in Pr$^{4+}$ systems.

Finally, to gain ligand-based information about Pr-4f/O-2p hybri-
dization, O $K$-edge XAS[38] were acquired. The spectra for **2-Pr** and **0-Pr**,
Fig. 3c,d, reveal strong features in the 532.8 to 536 eV range attributed

to excitations from the $1s$ shells of the ligand to states with Pr-5d and
O-2p character. These features are a measure of the $5d$-covalency of
the Pr−O bond and show that nominally unoccupied $5d$ orbitals take
part in covalent bonding[39]. The splitting of the $5d$ states is estimated to
be 3.67(11) eV in **2-Pr** and 3.61(4) eV in **0-Pr** and compares well with the
value calculated for PrO$_2$ ($\approx 3.6$ eV, see Fig. S3)[28]. Contributions from Pr-
$6sp$ states cannot be entirely neglected in the $5d$-driven region[40]. More
subtle features common to both **2-Pr** and **0-Pr** include pre-$5d$-edge
peaks at near $\approx 529$ and $\approx 530.7$ eV (labeled as $1s \rightarrow 4f$). These pre-edge
peaks are a signature of strong Pr-4f/O-2p hybridization in the ground
state ($|\psi_g\rangle$) because they reflect transitions from the O $1s$-core states to
$2p$-hole states of the oxygen in the narrow $4f$-dominated bands. These
low-energy pre-edge features are characteristic of Ln$^{4+}$ ions; if at all
present in spectra of Ln$^{3+}$ systems[39], they are quite weak. The inte-
grated intensities of the $1s \rightarrow 4f$ peaks is 3.4(1) and 2.5(1) larger for PrO$_2$
and **2-Pr**, respectively, than for **0-Pr**. This result indicates that **0-Pr** has
the least Pr-4f/O-2p hybridization in good accord with the Pr $M_{5,4}$ edge
spectra. Overall, the presence of pre-edge features in the O $K$-edge XAS
spectra confirms $4f$ covalency in the Pr-O bond and strongly indicates
ligand holes induced by Pr-4f/O-2p hybridization[40].

X-ray absorption provides unambiguous evidence for the presence
of $4f − 2p$ hybridization. If the $|4f^2 \underline{\nu}\rangle$ character is localized (as in a truly
multiconfigurational ground state), it would be reflected as CF excita-
tions corresponding to Pr$^{3+}$ (usually $\leq 50$ meV)[2]. It is evident that such CF
transitions from localized Pr$^{3+}$ are not observed in both neutron and IR

spectra. The alternative and the most reasonable explanation is that the $|4f^2 \underline{\nu}\rangle$ character is delocalized in the Pr-O bond as is the case for charge-transfer insulator like nickelates[41] and cuprates[42]. This framework is inconsistent with $\hat{\mathcal{H}}_{CF}^{Pr}$ described earlier, where the basis states are pure $4f$ functions. The hybridized orbitals are analogous to molecular orbitals with significant ligand contribution. Therefore, we use a modified $\hat{\mathcal{H}}_{CF}^{hyb} = \kappa^2 B_2^0 \hat{O}_2^0 + \kappa^4 B_4^0 \hat{O}_4^0 + \kappa^4 B_4^4 \hat{O}_4^4 + \kappa^6 B_6^0 \hat{O}_6^0 + \kappa^6 B_6^4 \hat{O}_6^4$ by including an orbital reduction factor[43,44] ($\kappa = 0.9$) which accounts for metal-ligand hybridization and yields similar results to the original model (see Supplementary Note 1).

## Discussion

Taken together, the present experiments point at Pr-$4f$/O-$2p$ hybridization as the essential microscopic mechanism behind the unusual electronic and magnetic properties of $Pr^{4+}$ systems. A qualitative understanding of Pr-O bonding is enabled by ab-initio calculations (CASPT2/CASSCF+SOC, see Methods and Supplementary Note 2). Considering an isolated $[PrO_6]^{8-}$ fragment with perfect $O_h$ symmetry, the $4f$ atomic orbitals (AOs) can be easily described in the $|m_l, m_s\rangle$ basis. Here, the CF splitting leads to three spin states: a $^2A_{2u}$ ground-state, and two excited $^2T_{1u}$ and $^2T_{2u}$ triply-degenerate states (Fig. 1a). The $^2A_{2u}$ state has a $\delta$-symmetry with respect to surrounding oxygens and thus remains strictly non-bonding. In contrast, the 4f AOs underlying the $^2T_{2u}$ and $^2T_{1u}$ states overlap with oxygen's $2p$ atomic orbitals leading to bonding and anti-bonding molecular orbitals (MOs) with $\pi$ and $\sigma + \pi$ character, respectively, about the Pr−O axes (See Fig. S14). When SOC is turned on, Table S10, the ground state corresponds to the admixture 58%$^2A_{2u}$ + 42%$^2T_{2u}$ (what identifies with the $E_{5/2u}$ term in the $O_h$ double-group symmetry). Departing from $O_h$ symmetry – as relevant for the PrO$_6$ distorted-octahedra of **0-Pr** and **2-Pr**−lifts the degeneracies of the $^2T_{1u}$ and $^2T_{2u}$ excited states (Tables S11, S12). However, regardless of distortion, the ground state in both **2-Pr** and **0-Pr** remains *solely* an admixture of $^2A_{2u}$ and $^2T_{2u}$ states.

The spectroscopy and thermomagnetic measurements are well explained by this model and calculations. For instance, the calculated $|\langle L_z \rangle / \langle S_z \rangle| \approx 1.8$ (See Table S12) is consistent with the XMCD data, and the small $Pr^{4+}$ magnetic moment can be attributed to self-compensating spin and orbital moments combined with an unusually small $|\langle L_z \rangle / \langle S_z \rangle|$ that signals a strong reduction of the orbital character in the original $\Gamma_7$ ground-state doublet. This framework naturally explains the O $K$ edge spectra of **2-Pr** and **0-Pr** through ligand holes induced by the formation of hybridized $T_{1u}2p_\sigma + T_{2u}2p_{\sigma+\pi}$ states. The model also explains why the largest hybridization is observed for PrO$_2$: the eight-, rather than six-, oxygen coordination environment allows the Pr $4f$ $a_{2u}$ orbital to covalently interact with the O $2p$ orbitals with $\sigma$ symmetry, thereby exhibiting enhanced $4f$-$2p$ hybridization[28]. The difference in $4f$-$2p$ hybridization between **2-Pr** and **0-Pr** likely comes from different point-group symmetries for the PrO$_6$ unit and the overall symmetry of the material. It is clear that $4f$-$2p$ hybridization can strongly influence single-ion energy scales such as the CF - this phenomenon is directly analogous to the behavior of $d$-block metals.

Beyond single-ion properties, Pr-$4f$/O-$2p$ hybridization leads to unusually large two-ion magnetic exchange interactions. For example, $J = 1.2$ meV has been reported by some of us on **2-Pr**[17]; a value 2.5 times larger than the typical scale of $J \approx 0.4$ meV observed for $4f^1$ or $4f^{13}$ systems such as KCeO$_2$ and NaYbO$_2$[24,45]. The Weiss constant of **1-Pr** of around $|\Theta_{CW}| = 7$ K is also large, especially considering the quasi-1D nature of this system. MO theory can be used to understand these exchange interactions, as shown in Fig. 4a. In the charge transfer limit ($U_{ff} \gg \Delta E_g$), the nearest-neighbor exchange interaction scales as $t_{pf}^4 / \Delta E_{pf}^3$, where $t_{pf}$ is the hopping integral between $4f$ and $2p$ orbitals, and $\Delta E_{pf}$ is their energy difference (i.e., the ligand to metal charge transfer (LMCT)). The enhancement of magnetic exchange in Ln$^{4+}$ compounds is likely primarily driven by the reduction of the charge transfer energy $\Delta E_{pf}$, as evidenced by e.g., calculations of $[CeCl_6]^{3-}$ (>5 eV) and $[CeCl_6]^{2-}$ (-3.3 eV)[46]. Large $t_{pf}$ hopping and small $\Delta_{pf}$ implies

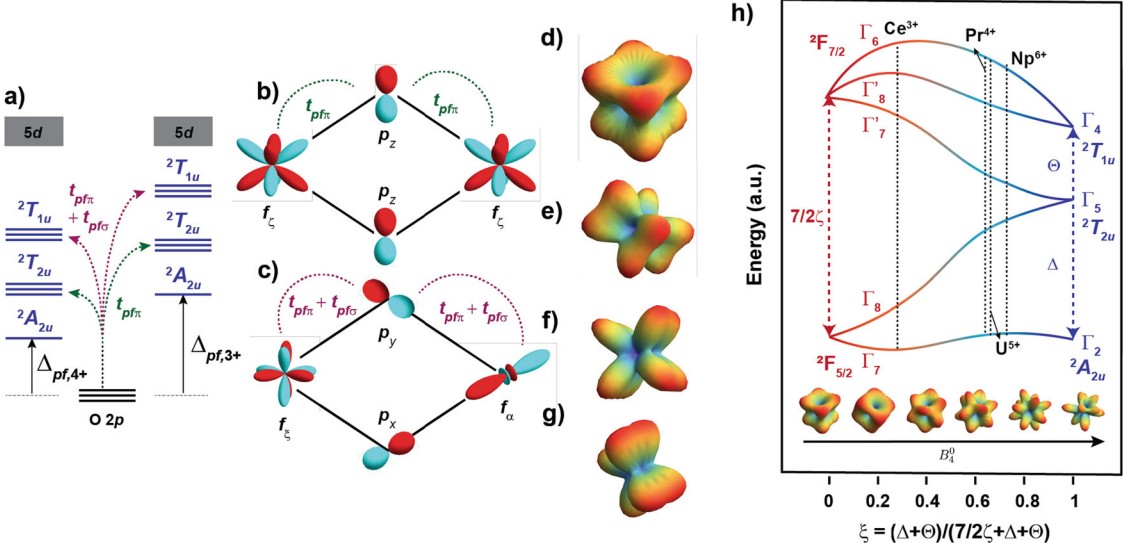

**Fig. 4 | Microscopic origin of anomalous properties of Pr⁴⁺ and a universal model for $f^1$ single-ions. a** Schematic of $p$ and $4f$ energy levels for Pr⁴⁺ and Ce³⁺. $t_{pf\pi}$ ($t_{pf(\pi+\sigma)}$) is the hopping integral between $p$ and $^2T_{2u}$ ($^2T_{1u}$) orbitals. The corresponding $pf$ charge transfer gap is indicated with $\Delta_{pf,4+} < \Delta_{pf,3+}$. **b** Sketch of the hopping processes between occupied $f_\zeta$ orbitals mediated by the $\pi$ interacting $2p$ orbitals analogous to $t_{2g} - p - t_{2g}$ hopping in $d^5$ systems. **c** Sketch of the hopping processes between occupied $f_\zeta$ and unoccupied $f_\alpha$ orbitals mediated by the $\pi + \sigma$ interacting $2p$ orbitals analogous to $t_{2g} - p - e_g$ hopping in $d^5$ systems. **d−g** Probability density of the ground state KD in ideal $\Gamma_7$, **2-Pr**, **1-Pr**, and **0-Pr**, respectively and shows the impact of mixing excited states in to the original $\Gamma_7$

doublet. **h** Schematic of the splitting of $f$ orbitals as a function of CF ($\Delta$ and $\theta$) relative to SOC ($\zeta$). The value of $\xi$ for Pr⁴⁺ was calculated from **0-Pr**, and the values for Ce³⁺, U⁵⁺, and Np⁶⁺ were obtained from ref. 49. Using this as a universal model for $f^1$ ions, Pr⁴⁺ is categorized together with the actinides, where the traditional Ln³⁺ picture breaks down. The figure also shows the evolution of the shape of the $\Gamma_7$ KD as a function of $B_4^0$ in the $\hat{\mathcal{H}}_{CF}^{O_h} = B_4^0 \hat{O}_4^0 + B_4^4 \hat{O}_4^4$, where $B_4^4 = 5B_4^0$. Increasing $B_4^0$ from -0 ($\zeta_{SOC} >> \Delta_{CF}$) yielding an almost perfect $\Gamma_7$ KD (leftmost figure) to -2000 ($\zeta_{SOC} << \Delta_{CF}$, rightmost figure) and the resultant drastic changes of the nature of the KD. The original nature of the $\Gamma_7$ KD is retained until the eigenvalue of the $\Gamma_8 \approx 75$ meV where $J_{eff} = 1/2$ limit still applies.

a large ligand-hole character, consistent with our O $K$-edge spectra. First-principles calculations on a binuclear $[Pr_2O_{10}]^{12-}$ embedded cluster model for **2-Pr** (see Supplementary Note 2) quantitatively confirm this picture. In the $S_{eff} = 1/2$ limit, the spin-singlet minus spin-triplet energy, which identifies with the Heisenberg exchange interaction, yields $J = 4.2$ meV (See Table S13, Fig. S16). Upon including SOC, the energy splitting among the four lowest levels is $\approx 1.5$ meV. To the extent that this energy separation approximates $J$, it is in good agreement with the measured value for **2-Pr** (1.2 meV). Therefore exchange interactions in $Pr^{4+}$ materials may change by an order of magnitude (0.3 to 4.2 meV) under changes of the ligand field (Fig. 1a). Similar effects have been observed in high-valent actinides, including $U^{5+}$ and $Np^{6+}$ [47]. Inspecting hopping pathways is also informative to comment on the Kitaev (AFM/FM) interactions proposed for **2-Pr** through the $T_{2u} - p - T_{1u}$ pathway (Fig. 4b) analogous to the $t_{2g} - p - e_g$ process in $d^5$ systems [16]. While this contribution is small in $d^5$ systems due to the large $t_{2g}$ to $e_g$ separation, it is proposed to be larger in $4f^1$-systems owing to the small CF energy scale [16]. However, as we have demonstrated, $Pr^{4+}$ systems exhibit a very large CF splitting (or in other words, the $J_{eff} = 1/2$ limit is not adequate), making the $T_{2u} - p - T_{1u}$ pathway energetically less favorable [48] than the $T_{2u} - p - T_{2u}$ pathway (Fig. 4c) responsible for the large Heisenberg AFM interaction.

Finally, and very importantly, the competition between CF and SOC energy scale in $Pr^{4+}$ systems resembles that in high-valent actinide systems such as $U^{5+}$ and $Np^{6+}$ for which a CF energy scale as large as $\approx 800$ meV is possible [49,50]. The chemistry and physics of high-valent actinides is further complicated by an extra competition between Coulomb repulsion, CF, and SOC [51,52] leading to the dual nature of $5f$ electrons [53]. In order to develop a universal description for $f^1$ single-ions, we argue that $Pr^{4+}$ systems can facilitate the study of the delicate balance of various competing interactions in the absence of competing Coulomb repulsion. To showcase that idea, we use the model established by ref. 50 where, the CF transitions for a $f^1$ ion in an ideal $O_h$ symmetry can be written as a function of two CF parameters $\Delta$ and $\theta$, and $\zeta_{SOC}$, as shown in Fig. 4f. Using this framework and our experimentally determined values for **0-Pr**, we calculate the parameter $\xi = \frac{\Delta + \theta}{7/2\zeta + \Delta + \theta} \approx 0.62$ for $Pr^{4+}$. When compared with other $f^1$ ions, including $Ce^{3+}$, $U^{5+}$, and $Np^{6+}$ (see Fig. 4h and refs. 49,54), it is evident that $Pr^{4+}$ lies closer to $U^{5+}$ than $Ce^{3+}$. Qualitatively, the observed trend can be generalized by a simple $f$-orbital bonding picture that puts hybridization with the ligands $np$-orbitals as the key microscopic phenomenon leading to an enhanced CF energy scale.

In summary, our work elucidated the mechanisms behind the anomalously large CF energy scale in $Pr^{4+}$ systems and discussed how exotic magnetic and electronic properties emerge as a result. The covalent character of the Pr–O bond plays a key role in determining the single-ion and macroscopic physics in $Pr^{4+}$ compounds, similar to familiar systems such as cuprates and nickelates. It is in sharp contrast to $Ln^{3+}$ systems for which, conventionally, the metal-ligand bond is described using an ionic picture. While the focus of this study has been oxides, $Pr^{4+}$ materials do exist as fluorides which require rigorous synthetic conditions using pure $F_2$ gas. The ability to stabilize Pr in the unusually high 4+ oxidation state demonstrates that there is a rich chemical space still to be explored beyond fluorides and oxides for designing new quantum materials, including mixed-anion materials. Our results define a strategy to design and control quantum materials by tuning the Pr–O covalency through site symmetry and ligand identity by moving to softer donors like S or Se. The inadequacy of the $J_{eff} = 1/2$ limit shows us how to change the fabric of spin-orbit entangled single-ion states to stabilize exotic exchange Hamiltonians or to develop universal models to understand the physics of high-valent actinides. $Pr^{4+}$-based systems offer the rare possibility to tune the hierarchy of single-ion energy scales, as well as the $p$ and $f$ hole density, which may be harnessed to design new correlated phenomena in quantum materials.

## Methods

### Material synthesis

$Na_2LnO_3$ (1-Ln, Ln = Ce, Pr) were synthesized using established procedures [18]. Polycrystalline powder samples of $Sr_2LnO_4$ (2-Ln, Ln = Ce, Pr) and $Li_8LnO_6$ (3-Ln, Ln = Pr, Tb) were synthesized using traditional solid-state methods. The samples were fired under a flow of $O_2$ in a tube furnace. The firing was performed at 1100 °C for 24 h. The samples were taken out of the quartz tubes in air and placed into the antechamber of the glovebox as quickly as possible in order to minimize contact with the ambient atmosphere (See SI for details).

### Experimental characterizations

**PXRD.** Laboratory powder X-ray diffraction (PXRD) was collected on a PANalytical X'Pert PRO Alpha-1 diffractometer with Cu $K\alpha$ source in reflection geometry equipped with a fixed divergence slit of 1/8", a convergence slit of 1/4" and a working radius of 240 mm. The samples were homogenized by finely grinding them inside the glovebox using an agate mortar for about -15 min. To avoid the exposure of the sample to atmospheric air, the PANalytical domed sample holder is equipped with stainless steel base and a polycarbonate dome with a 70% X-ray transmission. A $2\theta$ range of 5−85° was used with a scan speed of 5 s and a step size of 0.1.

**Physical property measurements.** The d.c. magnetic susceptibility measurements and isothermal magnetization measurements were made using a Quantum Design MPMS3. The sample was sealed in a plastic capsule on a low-background brass holder.

**Neutron-scattering measurements.** Inelastic neutron-scattering measurements were carried out on -8 g of polycrystalline samples of **1-Pr**, **1-Ce**, **0-Pr**, and **0-Tb** on the time-of-flight fine-resolution Fermi chopper spectrometer SEQUOIA, at the Spallation Neutron Source, ORNL [55,56]. The powder samples were enclosed in standard 15-mm diameter cylindrical annular aluminum cans (with 2 mm sample gap) under one atmosphere of $^4$He at room temperature. All four samples were cooled using a closed-cycle refrigerator reaching a base temperature of $T = 5$ K. Measurements were carried out using incident neutron energies $E_i = 300$, 500, 700 meV at $T = 5$ K. Background and sample holder contributions were measured using empty can measurements. The lattice phonon contributions for **0-Pr** and **1-Pr** were subtracted by measuring the analogous **0-Tb** and **1-Ce**, respectively.

**STXM O $K$ edge XAS.** STXM methodology was similar to that discussed previously [28]. In an argon-filled glovebox, samples for STXM measurement were prepared by pulverizing the powder compounds and transferring particles to $Si_3N_4$ windows. Second windows were placed over the samples to sandwich the particles, and the windows were sealed together with Hardman Double/Bubble epoxy. Single-energy images and O $K$-edge XAS spectra were acquired using the STXM instrument at the Canadian light source (CLS) spectro-microscopy beamline 10ID-1, operating in decay mode (250 to 150 mA, in a -0.5 atm He-filled chamber) at a base temperature of $T = 300$ K. The beamline uses photons from an elliptically polarizing undulator that delivers photons in the 130 to 2700 eV energy range to an entrance slit-less plane-grating monochromator. The maximum energy resolution $E/\Delta E$ was previously determined to be better than 7500, which is consistent with the observed standard deviation for spectral transitions of ±0.1 eV determined from the comparison of spectral features over multiple particles and beam runs. For these measurements, the X-ray beam was focused with a zone plate onto the sample, and the transmitted light was detected. The spot size and spectral resolution were determined from the characteristics of the 35 nm zone plate. Images at single energy were obtained by raster-scanning the sample and collecting transmitted monochromatic light as a function of the sample position. Spectra at particular regions of interest on the sample

image were extracted from the "stack", which is a collection of images recorded at multiple, closely spaced photon energies across the absorption edge. Dwell times used to acquire an image at a single photon energy were 2 ms per pixel and spectra were obtained using circularly polarized radiation. The incident beam intensity was measured through the sample-free region of the $Si_3N_4$ windows. In order to ensure that the spectra were in the linear regime of Beer–Lambert's law, particles with an absorption of less than 1.5 OD were used. High-quality spectra were obtained by averaging measurements from multiple independent particles, samples, and beam runs.

**STXM Pr $M_{5,4}$ edge XAS.** Measurements at the Pr $M_{5,4}$-edges were conducted using the STXM instrument at the Canadian Light Source (CLS) spectromicroscopy beamline 10ID-1, operating in top-off mode (250 mA, in a -0.5 atm He-filled chamber) at a base temperature of $T = 300$ K. The sample preparation and data acquisition methodology is the same as described above for the O $K$-edge measurements.

**Pr $M_{5,4}$ edge XMCD.** The XAS and XMCD measurements at Pr $M_{5,4}$-edges were conducted at beamline 4-ID-C of the Advanced Photon Source located at Argonne National Laboratory. XAS and XMCD spectra were collected simultaneously using total electron yield (TEY) and total fluorescence yield (TFY), respectively, with circularly polarized X-rays in a near normal (80°) configuration using a cryostat reaching a base temperature of $T = 20$ K. The applied field was along the beam direction and it defines the positive $Z$ direction. The data was obtained at both a zero field and an applied field of $\mu_O H = \pm 5$ T. The XMCD spectra were obtained point by point by subtracting right from left circular polarized XAS data. Measurements were taken for both positive and negative applied field directions, and then a difference of these two spectra XMCD = $\frac{1}{2}$ [XMCD($H_z > 0$) - XMCD($H_z < 0$)] was taken to eliminate polarization-dependent systematic errors. The TFY XAS data is identical to the STXM data described above. However, the TFY XMCD signal is weak and distorted by self-absorption effects. The TEY XAS data is similar to STXM data as well, except the high energy satellite peaks at both $M_{5,4}$ edges are weak and not as pronounced. Furthermore, the low-energy shoulder at the $M_4$ edge is more pronounced in TEY XAS than in both TFY and STXM. For discussions in the main text regarding $M_{5,4}$ edge isotropic XAS spectra, only the STXM data is discussed as it minimizes error due to self-absorption, saturation, and surface contamination. However, for our discussions with XMCD, we use the data collected in TEY mode. As noted in Fig. 3, the isotropic XAS was in the top panel and is measured in STXM mode, while the XMCD was at the bottom panel and is measured in TEY.

**Infrared magnetospectroscopy.** Broadband IR measurements were performed in the Voigt transmission configuration using a Bruker 80v Fourier-transform IR spectrometer. The incident IR light from a global source was guided to the top of the probe inside an evacuated beamline and then delivered to the bottom of the probe through brass light pipes. The sample was located in the middle of two confocal 90° off-axis parabolic mirrors mounted at the bottom of the probe. While the first mirror focuses the IR radiation on the sample, the second mirror collimates the radiation to the short light pipe with a 4K composite Si bolometer at the end. About 25 mg of the powder sample was mixed with KBr inside a glovebox in a 1:1 ratio. The resulting mixture was pressed into 3 mm pellets and was secured by a thin polypropylene adhesive film, and mounted on the brass plate with a clear aperture of 3 mm. The sample was placed at the center of a $\mu_0 H = 17.5$ T vertical bore superconducting magnet in a helium exchange gas environment, providing the sample temperature of about 5.5 K. IR transmission spectra were collected for 3 min at a fixed magnetic field, changing with 1 T step. All spectra obtained at different magnetic fields were normalized to the same reference spectrum, which is their mean, computed after removing the outlier points at each frequency. Such

normalization flattens those spectral features independent of magnetic fields and highlights those absorption peaks that shift as the magnetic field rises.

**Crystal field modeling of inelastic neutron scattering (INS)**
CF modeling was carried out using the truncated CF Hamiltonian $\hat{\mathcal{H}}_{CF} = B_2^0 \hat{O}_2^0 + B_4^0 \hat{O}_4^0 + B_4^4 \hat{O}_4^4 + B_6^0 \hat{O}_6^0 + B_6^4 \hat{O}_6^4$ where $B_n^m$ are the second, fourth, and sixth-order terms and $\hat{O}_m^n$ are the corresponding Stevens operator equivalents[57] for all three materials studied here. The Stevens operators are expressed in terms of $L$ and $L_z$. Although the true symmetries of $Pr^{4+}$ in each system require more parameters based on point-group symmetry, any mixing induced by these parameters would not induce any further loss of degeneracy and hence we choose to parameterize their effects $B_n^0$ and $B_n^4$ parameters. This approach was taken in order to minimize over-parametrization while fitting to experimental data. All Hamiltonian diagonalizations were performed using the PYCRYSTALFIELD package[58]. Fitting was carried out to a combination of eigen-energies extracted experimentally from INS and IRMS and to the temperature-dependent susceptibility data over $T > 40$ K in order to avoid short-range correlations present at lower temperatures. The final fit results are provided in Supplementary Table 3. The CF models were validated by calculating the isothermal magnetization at $T = 50$ K. The model calculation of $g$ values for the ground state wavefunction was compared to experimentally determined values from Curie–Weiss fits and first-principles calculations. See Supplementary Note 1 for a detailed description of the fitting procedure. The $f$ electron density plots were obtained using QUANTY[59] and plotted using Wolfram-Mathematica[60].

The $\Gamma_7$ KD in the $\Delta_{CF} << \zeta_{SOC}$ limit is written in the $|J, m_J\rangle$ basis as $\sin\theta|\frac{5}{2}, \pm\frac{5}{2}\rangle + \cos\theta|\frac{5}{2}, \mp\frac{3}{2}\rangle$, where $\sin\theta^2 \sim 1/6$. The same $\Gamma_7$ KD can be written in the $|m_l, m_s\rangle$ basis as $A|\mp 3, \pm\frac{1}{2}\rangle - B|\mp 2, \pm\frac{1}{2}\rangle + C|\pm 1, \pm\frac{1}{2}\rangle - D|\pm 2, \mp\frac{1}{2}\rangle$, where $A = 0.352$, $B = 0.215$, $C = 0.454$, $D = 0.79$, yielding $\alpha = \frac{A^2 + B^2}{C^2 + D^2} \approx 0.18$. The first two components of the $\Gamma_7$ KD in $|m_l, m_s\rangle$ ($m_l = \mp 3, \mp 2$) map onto a linear combination of the $|\frac{5}{2}, \pm\frac{5}{2}\rangle, |\frac{7}{2}, \pm\frac{5}{2}\rangle$, states in $|J, m_J\rangle$ basis, while the last components ($m_l = \pm 1, \pm 2$) map onto $|\frac{5}{2}, \pm\frac{3}{2}\rangle, |\frac{7}{2}, \pm\frac{3}{2}\rangle$ states. For the $\Gamma_7$ KD, given that $\Delta_{CF} << \zeta_{SOC}$, the contributions from the $J = \frac{7}{2}$ SOC manifold are negligible. As $\Delta_{CF} \sim \zeta_{SOC}$, non-negligible contributions from the $J = \frac{7}{2}$ SOC manifold enter the ground-state wavefunction making it impossible to deconvolute the individual contributions from each SOC manifold. Therefore, a better description of mixing can be obtained by looking at the ratios $\frac{A^2}{B^2}$ and $\frac{C^2}{D^2}$. Within this framework, irrespective of the symmetry at the metal center, for a six-coordinate system, the ground state wavefunction is always a linear combination of $m_l = \pm 1, \pm 2, \mp 3$ states. This derives from the $^2A_{2u} + ^2T_{2u}$ symmetry (in $O_h$ notation) as described in the main text and predicted by first-principles calculations. The introduction of intermediate coupling, changes only the relative mixtures of $m_l = \pm 1, \pm 2, \mp 3$ states and does not introduce any new admixture into the ground state wavefunction. The relative change in a mixture can be viewed as introducing $|\frac{7}{2}, \pm\frac{5}{2}\rangle$ and $\frac{7}{2}, \pm\frac{3}{2}\rangle$ states and increasing the amount of $|J, \pm\frac{5}{2}\rangle$ character in the ground state. This is clearly evident in the toy model established in Supplementary Note 1.

**Multiplet modeling of X-ray absorption spectroscopy (XAS)**
Multiplet calculations were implemented using the original code written by Cowan (ref. 32) and further developed by de Groot based on AIM. The multi-electron configuration in the ground and the final states was implemented using a charge-transfer methodology analogous to nickelates and cuprates. For all calculations, a Gaussian broadening of 0.45 eV was applied to account for instrumental broadening and Lorentzian broadening of 0.3 and 0.6 eV were applied

to the $M_5$ and $M_4$ edges, respectively. The model parameters had the following values for both 1-Pr and 3-Pr: $U_{ff}$ -14.1 eV, $U_{fc}$ -8.5 eV, $\zeta_{SOC}^{4f} \sim 0.12$ eV, and $\zeta_{SOC}^{3d} \sim 7.1$ eV where $U_{ff}$ and $U_{fc}$ are the $4f-4f$ Coulomb interaction and the core-hole potential acting on the $4f$ electron, respectively. In the limit of vanishing $V \to 0$, the difference between the two configurations in the ground state was $\Delta E_g = \epsilon_f - \epsilon_n = 2.0$ eV (1-Pr) and 3.0 eV (3-Pr), and $\Delta E_f = \epsilon_f - \epsilon_n + U_{ff} - U_{fc} = 0.5$ eV (1-Pr) and 1.5 eV (3-Pr), where $\epsilon_f$ and $\epsilon_n$ are one-electron energies of Pr $4f$ and O $2p$ levels and $V$ is the hybridization energy between atomic like localized $4f$ states and delocalized O $2p$ states which determines the mixing between the multi-electron configurations. Hybridization energy in the ground state ($V_g$) was set to 1.4 eV (1-Pr and 3-Pr), and final state ($V_f$) was set to 1.4 eV (1-Pr) and 1.8 eV (3-Pr).

## Ab-initio calculations

Without symmetry restrictions, wavefunction theory (WFT) calculations were performed within a commonly applied two-step spin-orbit coupled configuration-interaction formalism using OpenMolcas[61]. WFT calculations (See SI for more details) were performed on an isolated $PrO_6^{8-}$ cluster and on embedded cluster models for $Li_8PrO_6$ and $Na_2PrO_3$. Additional calculations were performed with a binuclear $Pr_2O_{10}^{12-}$ embedded cluster model of $Na_2PrO_3$. All geometries were extracted from the crystal structures and used without further optimization. A 40 Å sphere of atoms was generated from the crystal structure. An outer 32 Å sphere contained $Pr^{4+}$, $O^{2-}$, $Na^+/Li^+$ embedding point charges; the inner 8 Å sphere contained the $PrO_6^{8-}$ ion treated quantum-mechanically surrounded by $Pr^{4+}$, $O^{2-}$, $Na^+/Li^+$ pseudocharges represented by ab-initio model potentials (AIMPs). All atoms treated quantum-mechanically were modeled with all-electron atomic natural orbital relativistically contracted basis sets of valence triple-$\zeta$ quality (ANO-RCC-VTZP).

## Data availability

The datasets generated and/or analyzed during the current study have been deposited in the Figshare database under the accession code https://doi.org/10.6084/m9.figshare.22626946 and in the Zenodo repository under the accession code https://doi.org/10.5281/zenodo.7932208.

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

## Acknowledgements

We are thankful to Dr. Harry Lane for his insightful discussions. The work of A.R. and H.S.L.P. at Georgia Tech was supported by the Beckman Foundation as part of a Beckman Young Investigator Award to H.S.L.P. The work of J.K. and M.M. at Georgia Tech was supported by the National Science Foundation through Grant No. NSF-DMR-1750186 awarded to M.M. The work of Z.J. at Georgia Tech was supported by the US Department of Energy through Grant No. DE-FG02-07ER46451 awarded to Z.J. Some of this work was performed in part at the Materials Characterization Facility at Georgia Tech, which is jointly supported by the GT Institute for Materials and the Institute for Electronics and Nanotechnology, and is a member of the National Nanotechnology Coordinated Infrastructure supported by the National Science Foundation under Grant No. ECCS-2025462. This research used resources at the Spallation Neutron Source, a DOE Office of Science User Facility operated by the Oak Ridge National Laboratory. Use of the Advanced Photon Source at Argonne National Laboratory was supported by the US Department of Energy, Office of Science, Office of Basic Energy Sciences, under Contract DE-AC02-06CH11357. The infrared measurements were performed at the National High Magnetic Field Laboratory, which is supported by the National Science Foundation Cooperative Agreement No. DMR-1644779 and the State of Florida. The work of D.-C.S. and J.A. at the University at Buffalo was supported by the US Department of Energy, Office of Basic Energy Sciences, Heavy Element Chemistry program, under grant DESC0001136 awarded to J.A. D.-C.S. and J.A. thank the Center for Computational Research (CCR) at the University at Buffalo for providing computational resources. D.-C.S. received research funding from the European Union's Horizon 2020 Research and Innovation Program under Marie Sklodowska-Curie Grant Agreement No. 899546. D.-C.S. acknowledges infrastructure support provided through the RECENT AIR grant agreement MySMIS no. 127324. Work of J.A.B. and S.G.M at LBNL was supported by the Director, Office of Science, Office of Basic Energy Sciences, Division of Chemical Sciences, Geosciences, and Biosciences of the US Department of Energy (DOE) at LBNL under Contract No. DE-AC02-05CH11231. STXM research described in this paper was performed at the Canadian Light Source, which is supported by the Canada Foundation for Innovation, Natural Sciences and Engineering Research Council of Canada, the University of Saskatchewan, the Government of Saskatchewan, Western Economic Diversification Canada, the National Research Council Canada, and the Canadian Institutes of Health Research.

## Author contributions

A.R., M.M., and H.S.L.P. conceived the project, which was led by A.R. and H.S.L.P. A.R. synthesized the samples at Georgia Tech. A.R., J.K., A.I.K., and M.M. performed the neutron-scattering measurements. A.R. and J.K. analyzed the neutron-scattering data. A.R., M.M., and J.K. carried out thermomagnetic measurements. M.O. and Z.J. performed the IR measurements and analyzed the data. J.A.B. and S.G.M. performed STXM XAS measurements and normalized the data. A.R., J.W.F., and H.S.L.P. measured XMCD data. A.R. analyzed the X-ray scattering data. D.-C.S. and J.A. carried out the theoretical calculations and accompanying

analyses. A.R., M.M., and H.S.L.P. wrote the manuscript with input from all authors.

## Competing interests

The authors declare no competing interests.
