## [Peer Review File · Nature Communications]

Chemical Design of Electronic and Magnetic Energy Scales of Tetravalent Praseodymium in MaterialsREVIEWER COMMENTS

Reviewer #1 (Remarks to the Author):

The work of Ramanathan et al. focuses on oxide compounds containing the unusual formally 4+ valence state of Pr, which are investigated by a battery of experimental and theoretical methods. It is argued that the unusual Pr 4f1 configuration exhibits substantial hybridization with O 2p orbitals, generating an intrinsically covalent picture as opposed to the typical ionic description of 3+ lanthanides. This is shown to give rise to relatively huge crystal field splitting, transforming the energy hierarchy and accessing new magnetic states in Pr4+ systems. The authors thus argue for new vistas on quantum materials with Pr4+ ions.

I agree with the authors on the importance of these findings. This is, in fact, an impressive piece of work. The authors start from intriguing observations regarding Pr4+ in cuprates and then execute an unusually broad study. The latter encompasses polycrystalline synthesis in various dimensionalities, magnetometry, inelastic neutron spectroscopy, in-field spectroscopy, XAS/XMCD, and multiple forms of theoretical calculations. The experiments and theory are both sophisticated, and the way they are intertwined is unusually effective. In short, I find this to be an impressive piece of work with fascinating implications that I think is well suited to Nature Communications. The calculations at the end where it is concluded that Pr4+ is closer to U5+ than Ce3+ is particularly striking and strongly supports some of the authors contentions. I recommend acceptance but I would like the authors to think about some comments:

1. I think the impact of the work would increase further if, near the end of the paper, the authors can comment a bit on the scope for synthesis of new compounds in this class. Many physicists reading this paper will have little idea of the situation from the solid-state chemistry perspective. How big is this class of materials likely to be? How difficult are they to synthesize and design? I am not asking the authors to overstretch/speculate, rather just to comment on what they see as the scope for further exploration.

2. I think the authors sell short the connections to other systems in their discussion of PBCO cuprates. Consider for instance the following fascinating connections to other oxides/chalcogenides:

> The temperature-dependent Pr3+/Pr4+ transition known to occur in Pr-based cobaltites and recently stabilized to room temperature (see Chaturvedi et al., Nat Commun 13, 7774 (2022) and references within).

> The suspected role of Pr-O hybridization and covalent bonding in the structure of (Pr,Sr)CoO3 (Leighton et al., Phys Rev B 79, 214420 (2009)).

> The similar physics thought to occur in other rare-earth-based compounds in rock-salt structure (e.g., Jayaraman et al., Phys. Rev. Lett. 25, 1430 (1970)).

3. I found something a little muddy at the bottom of page 4, where it is stated that the 50 K M(H) data are reproduced by the theory. The M(T) data are, as well, right? Other than this, the paper is very clearly and precisely written.

4. There are a few minor errors and typos:

> p3. "octahedral in Li8PrO6". Should be "octahedra".

> p7. "ev". Should be "eV".

> p9. "microscopic phenomena". Should be "phenomenon".

> I also think the authors should rethink "in" in the title. In the current formulation, isn't "compounds" needed at the end?

Reviewer #2 (Remarks to the Author):

The reported study concerns the electronic and magnetic states of the $4f^1$ electronic configuration of praseodymium as found in certain insulating oxides. The authors have used neutron spectroscopy to determine the crystal-field-split low energy states of $4f^1$, and x-ray absorption spectroscopy and magnetic circular dichroism to investigate hybridisation effects. The work focuses on oxides containing Pr^{4+} ions in sites surrounded by oxygen octahedra with varying degrees of distortion. The overall conclusion is that $\text{Pr-}4f/\text{O-}2p$ hybridisation plays an essential role in the electronic structure of the studied materials.

The $4f^1$ configuration is known in great detail in compounds containing trivalent Ce, in which very strong hybridisation often occurs and leads to mixed valence and Kondo phenomena. Less work has been done on tetravalent Pr compounds, in which hybridisation is expected to be weaker due to the smaller radial extent of the f electron orbitals. So the present systematic study of three tetravalent Pr compounds with different structural dimensionalities is very welcome.

The manuscript is written clearly. The experiments and methods of data analysis are described very thoroughly, especially in the supplemental material, and I have confidence that the results are dependable and should be published. On the other hand, notions of mixed valence and hybridisation effects in tetravalent Pr oxides, especially PrO_2 , are actually quite well established. See, for example, various studies carried out in the 1980s by Kotani and co-workers, PRB 38, 3433 (1988) and J Phys.: Condens. Matter 9, 8155 (1987), as well as others PRB 36, 1745 & 1750 (1987). In this context, the main themes of the present work are not particularly novel, and I don't see a strong case for publication in a general science journal as opposed to a more specialized solid-state physics or chemistry journal. On the other hand, if the authors could reconcile the lack of any evidence for hybridisation in their neutron spectra with the significant amount of hybridisation required to model the x-ray spectra (see point 1 below) then in my view this would be a notable result appropriate for publication in Nature Communications.

My main comments are the following:

1. There is a lack of consistency between the interpretations of the neutron and x-ray spectroscopy. The x-ray spectra are interpreted in terms of a model which incorporates significant f-p hybridisation, resulting in 25-30% of Pr^{3+} mixed into the Pr^{4+} ground state. At the same time, the model used to interpret the neutron data (supported by fits to the bulk susceptibility) includes no hybridisation at all, being a single-ion model for the $4f^1$ configuration of Pr^{4+} in a crystal field. The degree of hybridisation implied by the analysis of the x-ray spectra would be expected to significantly influence the neutron spectra, e.g. by lifetime broadening the Pr^{4+} excitations, or by the presence of peaks due to localised Pr^{3+} . There doesn't seem to be any evidence for such effects in the neutron or IR spectra shown in Fig. 2. Final state effects are important in x-ray spectra, so perhaps it is necessary to look again at the way the x-ray spectra are modelled and question the extent to which hybridisation can really be determined from such data?

2. Related to first point, it would be very interesting to calibrate the neutron spectra in absolute units and to make a quantitative comparison with the intensity calculated in the same units from the single-ion model. If there is significant "missing" intensity this could indicate electronic levels which are not localised in pure $4f^1$ states. Would it be possible to calibrate the neutron intensity retrospectively?

3. In the Supplementary Material, Sections 3.1, 3.2 and 3.3, the discussion of different regimes for the crystal field interaction relative to spin-orbit coupling is slightly confusing because it suggests that the $|j, m_j\rangle$ basis is applicable when $\text{CF} \ll \text{SOC}$, and the $|m_L, m_S\rangle$ basis is applicable when $\text{CF} \gg \text{SOC}$. In fact, both are complete basis sets and can be used in any regime. I think what you mean to say is that the $|j, m_j\rangle$ basis is the more natural one for $\text{CF} \ll \text{SOC}$ because j is a good quantum number, whereas $|m_L, m_S\rangle$ is more natural for $\text{CF} \gg \text{SOC}$ where j is not a good quantum number. So the titles of these three sections might be better given as $\text{CF} \ll \text{SOC}$, CF

>> SOC and CF \sim SOC.

Some smaller points (in no particular order):

4. I believe that the Stevens operators used in the analysis of the neutron data here are expressed in terms of L and Lz. It would be helpful to confirm this.

5. The f electron density plots shown in the main article (Fig. 1, Fig. 4) are very insightful. It would be good to give some details of how these were calculated.

6. "3-Pr" is written in the legend of several of the supplementary figures. Should this be "2-Pr"?

7. Page 3, line 5 from top. Should J be Jex?

8. Page 5, last line. The branching ratio would be clearer if the denominator was in parentheses: $IM5/(IM5+IM4)$. Some details of how the branching ratios were determined from the experimental spectra would be helpful.

9. Fig. 4h. In the caption, it says that B40 was varied from 0 to \sim 2000, yet the fitted B40 values are all less than 1 (Table S3). Is this consistent with the position of Pr⁴⁺ shown on Fig. 4h?

10. The system Ba₂Pr(Ru,Ir)O₆ might be worth mentioning as part of the context for studying tetravalent Pr. It has a valence transition from Pr³⁺ to Pr⁴⁺ which is detected in neutron spectra through the CF transitions of the two valence states, see PRB 99, 184440 (2019).

Reviewer #3 (Remarks to the Author):

The submitted manuscript demonstrates the unique electronic structure of Pr⁴⁺ which, unlike other lanthanide ions exhibit a covalent character of the Pr-O bonds. This covalency enhances crystal-field effects that compete with the inherent spin-orbit coupling contribution found in a 4f¹ system. This makes the energy spacing in Pr⁴⁺-based complexes resemble that of high-valent actinides (eg. U⁵⁺).

The unexpected role of Pr⁴⁺ 4f electrons is demonstrated very nicely in this paper by the combination of metal and ligand edge XAS spectra, X-ray magnetic circular dichroism, Anderson-impurity model and ab initio calculations. The analysis of the data is very detailed and theoretically sounded. The discussion and arguments are very well formulated, making the paper easy to follow.

To summarize, this is a very interesting piece of work, with, I believe, a very new and important message to convey, as there has been so far very few studies on tetravalent lanthanides, and none highlighting a covalent interplay of 4f orbitals, so far believed as very ionic. This paper clearly opens the way to a new look at higher-valent lanthanide chemistry, and to potential applications to magnetism. I am positively content to see this work published in Nature Communications.

RESPONSE TO REVIEWERS' COMMENTS

Reviewer #1 (Remarks to the Author):

The work of Ramanathan et al. focuses on oxide compounds containing the unusual formally 4+ valence state of Pr, which are investigated by a battery of experimental and theoretical methods. It is argued that the unusual Pr 4f¹ configuration exhibits substantial hybridization with O 2p orbitals, generating an intrinsically covalent picture as opposed to the typical ionic description of 3+ lanthanides. This is shown to give rise to relatively huge crystal field splitting, transforming the energy hierarchy and accessing new magnetic states in Pr⁴⁺ systems. The authors thus argue for new vistas on quantum materials with Pr⁴⁺ ions.

I agree with the authors on the importance of these findings. This is, in fact, an impressive piece of work. The authors start from intriguing observations regarding Pr⁴⁺ in cuprates and then execute an unusually broad study. The latter encompasses polycrystalline synthesis in various dimensionalities, magnetometry, inelastic neutron spectroscopy, in-field spectroscopy, XAS/XMCD, and multiple forms of theoretical calculations. The experiments and theory are both sophisticated, and the way they are intertwined is unusually effective. In short, I find this to be an impressive piece of work with fascinating implications that I think is well suited to Nature Communications. The calculations at the end where it is concluded that Pr⁴⁺ is closer to U⁵⁺ than Ce³⁺ is particularly striking and strongly supports some of the authors contentions. I recommend acceptance but I would like the authors to think about some comments:

We thank the reviewer for providing valuable comments and positive feedback on the manuscript.

1. I think the impact of the work would increase further if, near the end of the paper, the authors can comment a bit on the scope for synthesis of new compounds in this class. Many physicists reading this paper will have little idea of the situation from the solid-state chemistry perspective. How big is this class of materials likely to be? How difficult are they to synthesize and design? I am not asking the authors to overstretch/speculate, rather just to comment on what they see as the scope for further exploration.

We agree this would be a valuable addition to the manuscript. Pr⁴⁺ materials are known only in oxides and fluorides. Given the high oxidation state, the metal center requires electro-negative ligands like F⁻ and O²⁻ to be stable. In a recent review, we have compiled a list of Pr⁴⁺ materials (DOI: [10.1039/D0DT01400A](https://doi.org/10.1039/D0DT01400A)), where we identify a total of 11 oxides and 21 fluorides. The table is provided here for reference.

Material	Structure	θ_{CW} (K)	μ_{eff} (μ_B)	Comments
Na ₂ PrO ₃	C2/c	-15	0.99	Entropy recovered - $\sim 0.71R\ln 2$
Li ₂ PrO ₃	Cmmm	-32	1.75	Entropy recovered - $\sim 0.71R\ln 2$
SrPrO ₃	Pbnm	-----	1.57	No magnetic ordering down to 4.2K
BaPrO ₃	Pnma	-12	0.7	Exhibits a series of phase transitions at HT ($\chi_0 = 6.9 \times 10^{-4}$ emu/mol)
Sr ₂ PrO ₄	Pbam	-7.3	1.2	Entropy recovered - $\sim R\ln 2$ ($\chi_0 = 6.61 \times 10^{-4} \mu_B$)
Li ₈ PrO ₆	R$\bar{3}m$	-----	0.505	Isolated octahedrons of Pr ⁴⁺ ($\chi_0 = 2.67 \times 10^{-4}$ emu/mol)
K ₂ PrO ₃	C2/c	-140	2.4	Isostructural to Na ₂ PrO ₃
Cs ₂ PrO ₃	Cmc2₁	-101	3.54	-----
Rb ₂ PrO ₃	C2/c	-----	-----	-----
NaPrF ₅	Rhombohedral			Isostructural to NaPuF ₅
KPrF ₅	-----	-----	-----	-----
CsPrF ₅	Rhombohedral	-37	2.38	Colorless,
Li ₂ PrF ₆	-----	-----	-----	Colorless, isostructural to Li ₂ ZrF ₆
Na ₂ PrF ₆	Immm	-70	2.25	Colorless
K ₂ PrF ₆	-----	-62	2.24	Colorless
Rb ₂ PrF ₆	Hexagonal	-44	2.18	Colorless, isostructural to Rb ₂ UF ₆
Cs ₂ PrF ₆	Hexagonal	-130	2.14	Colorless
Na ₃ PrF ₇	Cubic	-115	2.22	Colorless
K ₃ PrF ₇	-----	-----	-----	Colorless, isostructural to (NH ₄) ₃ ZrF ₇
Cs ₃ PrF ₇	Cubic	-97	2.21	Colorless
CdPrLi ₂ F ₈	-----	-----	-----	Colorless, Scheelite type
BaPrF ₆	-----	-----	-----	Colorless, isostructural to RbPaF ₆
PrF ₄	C2/c	-----	2.42	Colorless, isostructural to ZrF ₄
Rb ₂ CsPrF ₇	Cubic	-----	-----	Colorless
Cs ₂ RbPrF ₇	Cubic	-----	-----	Colorless
K ₂ RbPrF ₇	Cubic	-----	-----	Colorless
Rb ₂ KPrF ₇	Cubic	-----	-----	Colorless
Rb ₃ PrF ₇	Cubic	-----	-----	Colorless
CsRbKPrF ₇	Cubic	-----	-----	Colorless
Cs ₂ KPrF ₇	Cubic	-----	-----	Colorless
Rb ₂ Li ₁₄ Pr ₃ O	-----	-----	-----	Isostructural to K ₂ Li ₁₄ Pb ₃ O ₁₄
¹⁴ PrO ₂	Pnma	-105	2.32	Isostructural to CaF ₂

This is an exhaustive list of Pr⁴⁺ materials compiled from the ICSD and CCSD. We note here that, the reader should proceed with caution while looking at the information provided

here, since most of the materials described here, were synthesized in the 1960's and 70's and may be incompletely structurally characterized (as was the case for Na_2PrO_3). All the fluorides specified here were synthesized under rigorous conditions using F_2 gas. This makes the oxides a bit easier to synthesize since it can be done under O_2 or ozone (O_3/O_2 mixture). Furthermore, the alkali metal oxides of Pr^{4+} , were synthesized using A_2O (A = Li, Na, K, Cs, and Rb) as the starting material. These reactions do not proceed while using an indirect A_2O source like A_2CO_3 . However, A_2O oxides past Na are difficult to synthesize, requiring heating corresponding alkali metals under a controlled flow of O_2 , while Li_2O and Na_2O are commercially available.

We have included the following in the main text of the manuscript.

“While the focus of this study has been oxides, Pr^{4+} materials do exist as fluorides which requires a rigorous synthetic conditions using pure F_2 gas. The ability to stabilize Pr in the unusually high 4+ oxidation state demonstrates that there is a rich chemical space still to be explored beyond fluorides and oxides for designing new quantum materials including mixed-anion materials. Furthermore, changing the ligand to more softer donors like S or Se offers the ability to tune the spin-orbit coupled single-ion states in Pr^{4+} .”

Even though there are no reported examples of Pr^{4+} in selenides reported to date, we are pursuing this class and have preliminary evidence for Pr^{4+} in these phases.

2. I think the authors sell short the connections to other systems in their discussion of PBCO cuprates. Consider for instance the following fascinating connections to other oxides/chalcogenides:

We agree with the reviewer that there are some fascinating connections with other Pr containing materials parallel to PBCO. Initially, we did not want to oversell the unusual electronic structure of Pr^{4+} and thereby limiting ourselves to PBCO. We thank the reviewer for suggesting the suitable references which we believe will add more excitement to our interpretation and broadens the audience for our work.

> The temperature-dependent $\text{Pr}^{3+}/\text{Pr}^{4+}$ transition known to occur in Pr-based cobaltites and recently stabilized to room temperature (see Chaturvedi et al., Nat Commun 13, 7774 (2022) and references within).

This is a great reference showing the impact of $\text{Pr}^{3+}/\text{Pr}^{4+}$ valence transition on the structure-property relationship. This work was published as we were wrapping up our manuscript. This particular work shows the need to understand the single-ion electronic structure of Pr^{4+} and its impact on the macroscopic properties, and we believe our work fills in the gap. With this Pr^{4+} offers to ability to design and fabricate new functional materials with technological implications especially with oxide electronics accompanied by fundamental prospects of new quantum materials like nickelates and cuprates.

> The suspected role of Pr-O hybridization and covalent bonding in the structure of (Pr,Sr)CoO₃ (Leighton et al., Phys Rev B 79, 214420 (2009)).

This is again a great reference showing the importance of Pr-4*f*/O-2*p* hybridization and its impact on magneto-crystalline anisotropy.

> The similar physics thought to occur in other rare-earth-based compounds in rock-salt structure (e.g., Jayaraman et al., Phys. Rev. Lett. 25, 1430 (1970)).

This is an interesting example where samarium monochalcogenide shows a semiconductor-metal transition under pressure due to valence transition from Sm²⁺ to Sm³⁺ associated with a delocalization of the 4*f* electron to the 5*d* conduction band resulting in a change of electronic configuration from a 4*f*⁶ to a 4*f*⁵ 5*d*¹. Lanthanides in unusually low oxidation state like 2+ have been keen interest within our group particularly because of their ability to stabilize a 4*f*^{*n*}5*d*¹ configuration which leads to heavy-fermion/kondo behavior with the possibility for valence fluctuations. Here, in SmS, it has been proposed that, the 4*f*-ligand hybridization through delocalization via the 5*d* conduction band which makes it metallic under pressure. This is analogous to Pr⁴⁺ materials, where the hybridization is between Pr-4*f* and ligand-2*p* orbitals. This shows a clear dichotomy of how different types of hybridization can affect the bulk properties in a system.

We have added the following to the main text:

“The emergence of an insulating state in PBCO - a compound obtained by substituting Y by Pr in the high-*T_c* superconductor YBCO, and valence fluctuations driven metal-insulator transitions in Pr containing complex oxides – epitomizes such anomalous behavior. In PBCO, the significant Pr-4*f*/O-2*p* covalency (Fehrenbacher-Rice hybridization) drives a mixed-valent state for Pr ions that competes with Cu-3*d*/O-2*p* hybridization (Zhang-Rice) and dramatically suppresses superconductivity in favor of local magnetism. In Pr containing complex oxides like (Pr_{1-*y*}Y_{*y*})_{1-*x*}Ca_{*x*}CoO_{3-*δ*} and Pr_{1-*x*}Sr_{*x*}CoO₃, valence transition from Pr⁴⁺ to Pr³⁺ drives a spin state/metal-insulator transition, making them attractive for oxide electronics.”

And the following references have been added accordingly:

- 1) Chaturvedi, V. et al. Room-temperature valence transition in a strain-tuned perovskite oxide. Nature communications 13, 7774 (2022) (reference #14 in the main text)
- 2) Leighton, C. et al. Coupled structural/magnetocrystalline anisotropy transitions in the doped perovskite cobaltite $\text{Pr}_{1-x}\text{Sr}_x\text{CoO}_3$. Physical Review B 79, 214420 (2009) (reference #15 in the main text)

3. I found something a little muddy at the bottom of page 4, where it is stated that the 50 K M(H) data are reproduced by the theory. The M(T) data are, as well, right? Other than this, the paper is very clearly and precisely written.

We are sorry for the confusion. M(T) data in Figure 2, along with the eigen energies and degeneracies, is used to fit the crystal field Hamiltonian. The Stevens coefficients extracted from the fits are then used to calculate M(H) at T = 50 K. Reproducing the M(H) from the fit parameters supports our Hamiltonian.

4. There are a few minor errors and typos:
> p3. "octahedral in Li_8PrO_6 ". Should be "octahedra".

Changed

> p7. "ev". Should be "eV".

Changed

> p9. "microscopic phenomena". Should be "phenomenon".

Changed

> I also think the authors should rethink "in" in the title. In the current formulation, isn't "compounds" needed at the end?

We agree, the title is now "Chemical Design of Electronic and Magnetic Energy Scales of Tetravalent Praseodymium Materials."

Reviewer #2 (Remarks to the Author):

The reported study concerns the electronic and magnetic states of the $4f^1$ electronic configuration of praseodymium as found in certain insulating oxides. The authors have used neutron spectroscopy to determine the crystal-field-split low energy states of $4f^1$, and x-ray absorption spectroscopy and magnetic circular dichroism to investigate hybridisation effects. The work focuses on oxides containing Pr^{4+} ions in sites surrounded by oxygen octahedra with varying degrees of distortion. The overall conclusion is that $\text{Pr-}4f/\text{O-}2p$ hybridisation plays an essential role in the electronic structure of the studied materials.

We thank the reviewer for providing valuable comments on the manuscript.

The $4f^1$ configuration is known in great detail in compounds containing trivalent Ce, in which very strong hybridisation often occurs and leads to mixed valence and Kondo phenomena. Less work has been done on tetravalent Pr compounds, in which hybridisation is expected to be weaker due to the smaller radial extent of the f electron orbitals. So the present systematic study of three tetravalent Pr compounds with different structural dimensionalities is very welcome.

We agree that $4f^1$ configuration in Ce^{3+} has been extensively studied particularly regarding valence fluctuations, heavy fermion behavior and kondo phenomena. This is particularly limited to metallic systems where the hybridization is between 4f valence electrons and the conduction band 5d states. This essentially leads to a multiconfigurational approach to describe the ground state. However, given the systems studied in the current manuscript are insulators, the hybridization takes place between $\text{Pr-}4f/\text{O-}2p$ (ligand valence) orbitals which is different from the hybridization in kondo systems. This provides a clear dichotomy between the two different types of hybridization that drives the macroscopic properties of the system. Within this framework of insulating systems, we disagree with the reviewer that hybridization is expected to decrease as we go from Ce^{3+} to Pr^{4+} . Using ligand K-edge X-ray absorption spectroscopy measurements, it's been shown that metal-ligand hybridization increases in Pr^{4+} compared to both Ce^{3+} and Ce^{4+} (*J. Am. Chem. Soc.* 2017, 139, 49, 18052–18064, *J. Am. Chem. Soc.* 2015, 137, 7, 2506–2523).

The manuscript is written clearly. The experiments and methods of data analysis are described very thoroughly, especially in the supplemental material, and I have confidence that the results are dependable and should be published. On the other hand, notions of mixed valence and hybridisation effects in tetravalent Pr oxides, especially PrO_2 , are actually quite well established. See, for example, various studies carried out in the 1980s by Kotani and co-workers, PRB 38, 3433 (1988) and *J Phys.: Condens. Matter* 9, 8155

(1987), as well as others PRB 36, 1745 & 1750 (1987). In this context, the main themes of the present work are not particularly novel, and I don't see a strong case for publication in a general science journal as opposed to a more specialized solid-state physics or chemistry journal. On the other hand, if the authors could reconcile the lack of any evidence for hybridisation in their neutron spectra with the significant amount of hybridisation required to model the x-ray spectra (see point 1 below) then in my view this would be a notable result appropriate for publication in Nature Communications.

We thank the reviewer for providing these valuable references as context for our study. Our work was originally inspired by the beautiful work from Kotani, Kaindl, and co-workers. However, most of the XAS work showing evidence of hybridization has been limited to the binary oxide, PrO_2 , where the Pr^{4+} sits a pseudo cubic symmetry with eight oxygen coordination. We feel the publication of our results in a general science journal is essential because our work goes beyond binary oxide and broadens these concepts to lower symmetry materials that are more interesting from a quantum magnetism standpoint. In this current work, we have explored three different Pr^{4+} oxides, **0-Pr** (Li_8PrO_6), **1-Pr** (Sr_2PrO_4), and **2-Pr** (Na_2PrO_3), with different symmetries for the PrO_6 octahedra. This offers us the ability to understand the effect of symmetry on degree of metal-ligand hybridization, which in-turn impacts the single-ion properties. While we use X-ray scattering as a tool to identify metal-ligand hybridization in Pr^{4+} materials, the main theme of the present work is to establish the unique single-ion properties and macroscopic behavior of Pr^{4+} materials driven by metal-ligand hybridization. We show that the large crystal field (CF) splitting observed in Pr^{4+} competes with spin-orbit coupling (SOC) and requires an intermediate coupling scheme to describe the electronic structure which changes the expected Γ_7 doublet ground state. This indicates that the ionic SOC $>$ CF limit used to describe the trivalent lanthanide is inadequate as the paradigm shifts towards CF $>$ SOC. With this, we show that Pr^{4+} exhibits unique single-ion properties with a very small g value of ~ 0.8 due to self-compensating orbital and spin moments with $\mu_{\text{orb}}/\mu_{\text{spin}} = 2$ when compared to Ce^{3+} with $\mu_{\text{orb}}/\mu_{\text{spin}} = 8$. Beyond the single-ion properties, we also show the unusually large magnetic super-exchange interactions in Pr^{4+} materials due to metal-ligand hybridization. With this, we stipulate hopping pathways and provide rationale for the nature of exchange interactions in **2-Pr**. And finally, we build a cohesive model for f^1 systems, where show that the electronic structure of Pr^{4+} resembles high-valent actinide systems such as U^{5+} and Np^{6+} rather than its trivalent counterpart Ce^{3+} . The latter conclusion is important to build cohesive models of bonding across the f-element series.

Besides establishing the electronic structure of Pr^{4+} , Our results offer novel strategies to design and control quantum materials by tuning the Pr–O covalency through site symmetry and ligand identity potentially by moving to softer donors like S or Se or mixed-anion materials (i.e. $\text{O}^{2-}/\text{F}^{1-}$). The covalent character of the Pr–O bond plays a key role in determining the single-ion and macroscopic physics in Pr^{4+} compounds, like familiar systems such cuprates and nickelates, which is in sharp contrast to Ln^{3+} systems for which, conventionally, the metal-ligand bond is described using an ionic picture. The

key takeaway is that Pr⁴⁺ materials change the fundamental understanding of traditional trivalent lanthanides, which can be exploited to design new correlated phenomena in quantum materials. The spectroscopic resemblance between Pr⁴⁺ and high valent transition metal systems like nickelates and cuprates is striking and calls for more work to be done with high valent lanthanide systems to deconvolute the electronic structure. As pointed out by reviewer 1, the necessity to understand the electronic structure of Pr⁴⁺ is essential to design new materials with technical implications like oxide-based electronics by harnessing the valence fluctuations/spin-state switching. We hope this work would bring together a wide audience from a combination of solid-state and molecular chemistry, condensed matter physics, and materials communities and hence, we believe a general science journal like *Nature Communications* is the appropriate forum in contrast to more specialized physics/chemistry journals.

My main comments are the following:

1. There is a lack of consistency between the interpretations of the neutron and x-ray spectroscopy. The x-ray spectra are interpreted in terms of a model which incorporates significant f-p hybridisation, resulting in 25-30% of Pr³⁺ mixed into the Pr⁴⁺ ground state. At the same time, the model used to interpret the neutron data (supported by fits to the bulk susceptibility) includes no hybridisation at all, being a single-ion model for the 4f¹ configuration of Pr⁴⁺ in a crystal field. The degree of hybridisation implied by the analysis of the x-ray spectra would be expected to significantly influence the neutron spectra, e.g. by lifetime broadening the Pr⁴⁺ excitations, or by the presence of peaks due to localised Pr³⁺. There doesn't seem to be any evidence for such effects in the neutron or IR spectra shown in Fig. 2. Final state effects are important in x-ray spectra, so perhaps it is necessary to look again at the way the x-ray spectra are modelled and question the extent to which hybridisation can really be determined from such data?

We thank the reviewer for pointing this out. We do agree with the reviewer that, while X-ray scattering measurements show 4f-2p hybridization, our neutron scattering interpretation did not include hybridization (see below response to point 2: we have updated the analysis with a hybridization model). This important point has been a subject of numerous discussion within our collaboration and we decided to resolve it as follows. Neutron scattering does not necessarily disprove the presence of 4f-2p hybridization but X-ray scattering provides unambiguous evidence for the presence of 4f-2p hybridization. If the 4f²L̄ character is localized (as in truly a multiconfigurational ground state), it would be selected as crystal field excitations in the neutron data. Pr³⁺ usually has CF transitions < 50 meV (*Nat Commun* **4**, 1934 (2013)). It is evident that we do not see such CF transitions from localized Pr³⁺ in both neutron and IR spectra. The alternative and the most reasonable explanation is that the 4f²L̄ character is delocalized in the Pr-O bond as is the case for charge-transfer insulator like nickelates (*Journal of Physics: Condensed Matter* **9.8** (1997): 1679.) and cuprates (*Nature Phys* **5**, 867–872 (2009)). In this

framework, the hybridization would show up as missing intensity in the INS data when integrated in absolute units (discussed further in point two) and through the neutron scattering form factor.

We do agree with the reviewer that final state effects play a key role in X-ray absorption measurements. Our configuration-interactions (CI) calculations based on the Anderson impurity model do take in to account the final state effects (see methods section in the main text). Such CI modeling have proven worthy in the transition metal systems where there is significant metal-ligand hybridization (Phys. Rev. Lett. **68**, 2543). That being said, we do think that further advanced modeling is necessary to elucidate the electronic structure of these materials. We point the reviewer towards recent advances in interpreting X-ray absorption data (*Chemistry—A European Journal* 27.25 (2021): 7239-7251, *Physical Chemistry Chemical Physics* 24.18 (2022): 10745-10756). As evidenced in these papers, most of the work has been limited to CeO₂ where the Ce⁴⁺ is the metal center with 4f⁰ electronic configuration. It is essential to note here that, work beyond CeO₂ to other 4+ lanthanides has been limited, possibly due to the lack of model systems. Our work expands this sample space beyond the binary oxides like PrO₂ which should pave way for advanced theoretical modeling of the XAS data. However, such advanced modeling is beyond the scope of this work and we believe CI calculations provide sufficient evidence for metal-ligand hybridization in these materials.

The reviewer questions the degree of 4f-2p hybridization that can be determined from X-ray data. We agree that the value obtained is not a direct quantitative measure of covalency in the Pr-O due to final state effects and core-hole lifetimes (amongst other factors). This makes it difficult to provide a *quantitative* comparison of covalency between systems. However, with the series of systems studied here, it is reasonable to provide conclusions on the degree of hybridization based on relative measure of covalency. And hence in the main text, we report the relative integrated intensities of the 1s → 4f peaks rather than the absolute intensities. We note that, the XAS data was collected using the same method (STXM) for all three systems discussed here and PrO₂. STXM has been proven to be a more accurate method of measurement, particularly for soft X-ray absorption experiments, than fluorescence yield, with minimal self-absorption (*Chem. Rev.* 2020, 120, 9, 4056–4110). Quantitative measure of covalency from XAS measurements is further complicated with the lack of an intensity standard, particularly for O K-edge (*J. Am. Chem. Soc.* 2017, 139, 49, 18052–18064, *Chem. Rev.* 2020, 120, 9, 4056–4110). Recent work from Ed Solomon, Stefan Minasian, amongst others have been significant progress towards establishing an intensity standard for O K-edge XAS, making the technique as quantitative as possible (*J. Am. Chem. Soc.* 2017, 139, 49, 18052–18064,). However, within the context of this current work, the key takeaway is the presence of the pre-edge peak in O K-edge XAS which is a clear indication of 4f-2p hybridization and the tunability of the corresponding relative intensity with symmetry.

Confidential Data: Redacted

Confidential Data: Request Redacted

2. Related to first point, it would be very interesting to calibrate the neutron spectra in absolute units and to make a quantitative comparison with the intensity calculated in the same units from the single-ion model. If there is significant “missing” intensity this could indicate electronic levels which are not localised in pure $4f^1$ states. Would it be possible to calibrate the neutron intensity retrospectively?

This is a great point. The “missing” intensity problem was first answered in the canonical cuprates Sr_2CuO_3 (*Nature Phys* **5**, 867–872 (2009)). We agree this would be a great way of showing delocalized electronic levels given the resemblance of Pr^{4+} to cuprates. There are two ways to normalize our neutron scattering data. The first one relies on the measurement of a monochromatic vanadium standard (10% scatterer). Because the experiment was performed during the pandemic (and also given the general practices at Oak Ridge National Laboratory), we do not have access to a monochromatic vanadium run for these experiments. As ORNL’s Spallation Neutron Source is now undergoing a long 1+ year shutdown, we do not have a way to collect such normalization files at the moment.

Another approach is to self-normalize the data to elastic Bragg scattering. However, our materials suffer from significant hydrogen recoil line and OH vibrational modes which appears as an elastic incoherent scattering, dispersive background and flat modes in the data. The very oxidizing nature of Pr^{4+} makes them air and water sensitive. We believe the hydrogen recoil comes from either due to surface reaction to water or from the starting materials. Although we took immense care in the preparation of these samples and the transfer of samples to cans adequate for neutron scattering, our analysis of the starting materials suggest that the alkali metal oxides (Li_2O and Na_2O) are only 97% pure (highest grade commercially available) and contain 3% hydroxide impurities (confirmed used powder X-ray diffraction). Given such backgrounds, calculating the intensity from the neutron data would give us large uncertainties and will not be ideal for the proposed analysis. Furthermore, to address the missing intensity, high resolution data with single crystals is required, as shown for Sr_2CuO_3 . Given the oxidizing nature of these Pr^{4+} materials, synthesis of single crystals large enough for INS is not easy. **0-Pr** and **2-Pr** have alkali metals and hence often lead to vacancies under crystal growing conditions. We have attempted to grow single crystals of **1-Pr** using the laser heating czochralski method at the PARADIM facility (<https://www.paradim.org/>). However, at very high temperatures (2500°C), the sample melts incongruently leading to the formation of SrPrO_3 (Fig. 2a). Therefore, we plan on moving on to other Pr^{4+} materials where we can grow large enough single crystals like fluoride materials. Given the “missing” intensity problem has not yet been observed/resolved in Ln^{4+} materials, this calls for its own manuscript analogous to Sr_2CuO_3 . The analysis and results presented here provide ample motivation to tackle the significant chemical technical challenge – one we hope to rise to in the future. However, this study is beyond the scope of the current work, and is not necessary to support the conclusions made in this work.

Fig 2. (a) Powder X-ray diffraction of Sr_2PrO_4 after attempt of crystal growth at higher temperatures. (b) Powder X-ray diffraction of the starting material Na_2O showing impurities from NaOH .

However, in order include hybridization, we develop an alternative CF model by including an orbital reduction factor (κ). The use of orbital reduction has proven to be an effective method to include metal-ligand hybridization (*Nat Commun* **7**, 13773 (2016)). Within this framework, we rewrite the CF Hamiltonian as

$$H_{\text{CEF}} = \kappa^2 B_2^0 O_2^0 + \kappa^4 B_4^0 O_4^0 + \kappa^4 B_4^4 O_4^4 + \kappa^6 B_6^0 O_6^0 + \kappa^6 B_6^4 O_6^4$$

We have redone the analysis with the new Hamiltonian and the results are provided here.

	0-Pr	0-Pr (new)	1-Pr	1-Pr (new)	2-Pr	2-Pr (new)
B_2^0	1.3	1.04	-12.43	-17.11	-5.26	-7.97
B_4^0	0.54	0.45	0.76	1.12	0.76	0.54
B_4^4	2.29	2.81	3.43	5.31	3.43	2.13
B_6^0	-0.007	-0.001	-0.001	-0.008	-0.003	-0.08
B_6^4	0.11	0.13	0.45	0.8	0.112	0.20
ζ_{SOC}	112	100	112	100	112	100
g_{av}	~ 0.9	~ 0.9	~ 1.1	~ 1.1	~ 1.1	~ 1.1

A	0.241	0.336	0.428	0.408	0.407	0.413
B	0.331	0.315	0.293	0.263	0.331	0.298
C	0.363	0.409	0.344	0.338	0.351	0.370
D	0.837	0.788	0.783	0.806	0.776	0.777

Table 1. Fit parameters with the revised model compared with the model in original manuscript. A, B, C, and D correspond to coefficients of the wavefunction as described in the main text.

The inclusion of the revised Hamiltonian with hybridization does not change the conclusions of the current work. The revised model has been added to the SI as section 3.8. We note here that the κ value is 0.9 for all three materials irrespective of their different degree of hybridizations. It is important to remember that the different characterization suites used in the current work provide us with different information about the hybridization in the system (*M*-edge XAS, *O* *K*-edge XAS, *M*-edge XMCD, INS, and IR). Every technique has its pros and cons and therefore the information obtained from all of the experiments combined together helps us to build a cohesive model to understand covalency in the Pr-O, which is not possible with just one technique.

We have included the following in the main text:

“After establishing hybridization in the Pr-O bond using X-ray absorption, we turn back to CF model which assumed an ionic picture. A new CF model by including an orbital reduction factor (κ) was also employed to account for Pr-4*f*/O-2*p* hybridization and yields similar results to the original model.”

We have included a new section 3.8 in the SI to show the revised model.

3. In the Supplementary Material, Sections 3.1, 3.2 and 3.3, the discussion of different regimes for the crystal field interaction relative to spin-orbit coupling is slightly confusing because it suggests that the $|j, m_j\rangle$ basis is applicable when $CF \ll SOC$, and the $|m_L, m_S\rangle$ basis is applicable when $CF \gg SOC$. In fact, both are complete basis sets and can be used in any regime. I think what you mean to say is that the $|j, m_j\rangle$ basis is the more natural one for $CF \ll SOC$ because j is a good quantum number, whereas $|m_L, m_S\rangle$ is more natural for $CF \gg SOC$ where j is not a good quantum number. So, the titles of these three sections might be better given as $CF \ll SOC$, $CF \gg SOC$ and $CF \sim SOC$.

We acknowledge the reviewers concerns and we agree that both basis are suitable for any regime. The following changes have been made to the SI, the title for sections S3.1, S3.2, and S3.3 have been changed to

“S3.1 - CF splitting of Pr4+ in $\zeta_{\text{SOC}} \gg \Delta_{\text{CF}}$ regime.

S3.2 - CF splitting of Pr4+ in $\zeta_{\text{SOC}} \ll \Delta_{\text{CF}}$ regime.

S3.3 - CF splitting of Pr4+ in $\zeta_{\text{SOC}} \sim \Delta_{\text{CF}}$ regime.”

On this subject, recent progress in the chemistry community have been made in describing intermediate coupling regimes as described for Pr4+ (*J. Chem. Phys.* **2022**, 064112). We believe Pr4+ materials studied here would help improve such new theories and help us better understand the electronic structure of the actinides.

Some smaller points (in no particular order):

4. I believe that the Stevens operators used in the analysis of the neutron data here are expressed in terms of L and Lz. It would be helpful to confirm this.

Yes, the Stevens operators used in the analysis are expressed in terms of L and Lz. To make things clear, the following has been added to the methods section of the main text:

“The Stevens operators are expressed in terms of L and Lz.”

5. The f electron density plots shown in the main article (Fig. 1, Fig. 4) are very insightful. It would be good to give some details of how these were calculated.

Yes, we thought the electron density plots would be insightful. Thank you for your positive feedback. The plots were made using scripts written in Wolfram Mathematica. The spherical harmonics for the f orbitals were obtained from QUANTY. The following has been added to the methods section of the main text:

“The f electron density plots were obtained using QUANTY and plotted using Wolfram Mathematica.”

6. “3-Pr” is written in the legend of several of the supplementary figures. Should this be “2-Pr”?

Thank you for pointing this out. We apologize for the confusion. All incorrect labelling has been corrected in the SI.

7. Page 3, line 5 from top. Should J be J_{ex} ?

Yes, that’s correct. We have made the following change:

“The most noticeable consequences for Na_2PrO_3 are the low effective magnetic moment with $g \approx 1$ and the surprisingly large $J_{ex} \approx 1$ meV exchange interactions.”

8. Page 5, last line. The branching ratio would be clearer if the denominator was in parentheses: $\text{IM5}/(\text{IM5}+\text{IM4})$. Some details of how the branching ratios were determined from the experimental spectra would be helpful.

Parentheses has been added to the denominator. The branching ratios were calculated by integrating are under the pseudo-voigt functions used to fit the *M*-edge STXM data (Fig. S5).

9. Fig. 4h. In the caption, it says that B40 was varied from 0 to ~2000, yet the fitted B40 values are all less than 1 (Table S3). Is this consistent with the position of Pr⁴⁺ shown on Fig. 4h?

The B40 values correspond only to the electron density plots and not the actual f^1 model shown in the figure. The figures and the model do not have a one-one correspondence. The figures are just to show the trend of the electron density plots as the paradigm shifts. For instance, in Fig. S9, it is evident that a B40 value of ~1 is sufficient to get a $\xi = 0.72$ (in a perfect O_h symmetry, where $\Delta + \theta \sim 1300$ meV). We should note here that, one can never reach $\xi = 1$ limit since SOC is inherent to the ion. If B40 ~1 is sufficient to get $\xi = 0.72$, the values obtained from our fits (both ionic and hybridized model) are reasonable and justifies the position of Pr⁴⁺ in the f^1 model. It should also be noted here that the model is established for perfect O_h systems, and hence we chose 0-Pr as it is closer to a perfect O_h symmetry than the other two materials (evident from the small B20 value obtained for 0-Pr). B40 value of 2000 was chosen just to implement a unreasonable value for CF and to show the evolution of electron density in the $\xi = 1$ limit. One can think of the f^1 model as being on a logarithmic scale of B40 scale (just to provide a frame of reference). Again we would like to reiterate that the B40 values correspond

only to the electron density plots and are not a one to one correspondence with the f^1 model.

10. The system $\text{Ba}_2\text{Pr}(\text{Ru},\text{Ir})\text{O}_6$ might be worth mentioning as part of the context for studying tetravalent Pr. It has a valence transition from Pr^{3+} to Pr^{4+} which is detected in neutron spectra through the CF transitions of the two valence states, see PRB 99, 184440 (2019).

Yes, this is a great reference and has been included in the main text as reference #13.

Reviewer #3 (Remarks to the Author):

The submitted manuscript demonstrates the unique electronic structure of Pr^{4+} which, unlike other lanthanide ions exhibit a covalent character of the Pr-O bonds. This covalency enhances crystal-field effects that compete with the inherent spin-orbit coupling contribution found in a $4f^1$ system. This makes the energy spacing in Pr^{4+} -based complexes resemble that of high-valent actinides (eg. U^{5+}).

The unexpected role of Pr^{4+} 4f electrons is demonstrated very nicely in this paper by the combination of metal and ligand edge XAS spectra, X-ray magnetic circular dichroism, Anderson-impurity model and ab initio calculations. The analysis of the data is very detailed and theoretically sounded. The discussion and arguments are very well formulated, making the paper easy to follow.

To summarize, this is a very interesting piece of work, with, I believe, a very new and important message to convey, as there has been so far very few studies on tetravalent lanthanides, and none highlighting a covalent interplay of 4f orbitals, so far believed as very ionic. This paper clearly opens the way to a new look at higher-valent lanthanide chemistry, and to potential applications to magnetism. I am positively content to see this work published in Nature Communications.

We thank the reviewer for taking their time to review the manuscript and appreciate the positive feedback.

REVIEWERS' COMMENTS

Reviewer #1 (Remarks to the Author):

The authors responses to my points (and those of others as far as I can tell) are complete and satisfactory. The ensuing changes to the manuscript are generally good. I support publication at this stage.

RESPONSE TO REVIEWERS' COMMENTS

Reviewer #1 (Remarks to the Author):

The authors responses to my points (and those of others as far as I can tell) are complete and satisfactory. The ensuing changes to the manuscript are generally good. I support publication at this stage.

We thank the reviewer for providing valuable comments and their support for the manuscript.

Reviewer #2 (Remarks to the Author):

The authors have given detailed consideration to the reviewers' suggestions and have made changes which improve the manuscript. I am happy with their responses to my comments. I have a couple of follow-up comments to my first point which was about Pr-4f/O-2p hybridization, and which for me is the most interesting issue raised by the work.

We thank the reviewer for providing valuable comments and positive feedback on the manuscript.

First, it would be helpful if the authors could add some of the points they made in their rebuttal to the manuscript. For example, the point about their neutron and IR spectra ruling out a stable Pr³⁺ state, and the implication that the ligand hole is most likely located in the Pr-O bond. Second, relating to the authors new analysis of the inelastic neutron scattering data with a single-ion model that includes a phenomenological parameter kappa, it would be helpful to non-specialists if the authors could add some text to explain why the single-ion Hamiltonian modified with the kappa parameter is equivalent to the hybridization picture used to describe the X-ray spectra, in which the ground state wavefunction is taken to be a linear combination of $|4f^1\rangle$ and $|4f^2v\rangle$.

The following has been added to main text:

“X-ray absorption provides unambiguous evidence for the presence of 4f-2p hybridization. If the $4f^2\bar{L}$ character is localized (as in truly a multiconfigurational ground state), it would be reflected as CF excitations corresponding to Pr³⁺ (usually < 50 meV) (*Nat Commun* **4**, 1934 (2013)). It is evident that such CF transitions from localized Pr³⁺ are not observed in both neutron and IR spectra. The alternative and the most reasonable explanation is that the $4f^2\bar{L}$ character is delocalized in the Pr-O bond as is the case for charge-transfer insulator like nickelates (*Journal of Physics: Condensed Matter* **9.8** (1997): 1679.) and cuprates (*Nature Phys* **5**, 867–872 (2009)). This framework is inconsistent with \hat{H}_{CF}^{Pr} described earlier, where the basis states are pure 4f functions. The hybridized orbitals are analogous to molecular orbitals with significant ligand contribution. Therefore, we use a modified $\hat{H}_{CF}^{hyb} = \kappa^2 B_2^0 \hat{O}_2^0 + \kappa^4 B_4^0 \hat{O}_4^0 + \kappa^4 B_4^4 \hat{O}_4^4 + \kappa^6 B_6^0 \hat{O}_6^0 + \kappa^6 B_6^4 \hat{O}_6^4$ by including an

orbital reduction factor ($\kappa = 0.9$) which accounts for metal-ligand hybridization and yields similar results to the original model (see Suppl. Sec. 3.8).”

Apart from that, I would be happy to see the manuscript published in Nature Communications.

We thank the reviewer for the support.